# Structural basis for regulated assembly of the mitochondrial fission GTPase Drp1

Kristy Rochon[1], Brianna L. Bauer[1], Nathaniel A. Roethler[1], Yuli Buckley[1], Chih-Chia Su [1], Wei Huang [1], Rajesh Ramachandran [2,3], Maria S. K. Stoll[1,4], Edward W. Yu [1,2], Derek J. Taylor [1,2] & Jason A. Mears [1,2,4] ✉

Mitochondrial fission is a critical cellular event to maintain organelle function. This multistep process is initiated by the enhanced recruitment and oligomerization of dynamin-related protein 1 (Drp1) at the surface of mitochondria. As such, Drp1 is essential for inducing mitochondrial division in mammalian cells, and homologous proteins are found in all eukaryotes. As a member of the dynamin superfamily of proteins (DSPs), controlled Drp1 self-assembly into large helical polymers stimulates its GTPase activity to promote membrane constriction. Still, little is known about the mechanisms that regulate correct spatial and temporal assembly of the fission machinery. Here we present a cryo-EM structure of a full-length Drp1 dimer in an auto-inhibited state. This dimer reveals two key conformational rearrangements that must be unlocked through intramolecular rearrangements to achieve the assembly-competent state observed in previous structures. This structural insight provides understanding into the mechanism for regulated self-assembly of the mitochondrial fission machinery.

Dynamin superfamily proteins (DSPs) are a group of large, multi-domain GTPase proteins with a conserved catalytic domain and a stalk capable of driving self-assembly of oligomers and helical polymers. Found in bacteria and eukaryotic cells, their primary function is membrane remodeling, as distinct family members play important roles in vesicle budding and organelle fission and fusion. Given their important and varied roles, structural studies have sought to characterize DSP domain organization and contribution to membrane remodeling. In part, this was pursued using cryo-EM to examine DSP interactions on membrane templates or in complex with partner proteins in helical assemblies or filaments[1–4]. Through considerable effort, crystal structures were determined for individual DSP domains[5–7] and then for full-length proteins[8,9]. DSPs form insoluble assemblies at the higher concentrations required for crystallization, so several mutations were introduced to improve solubility[9–11].

Dynamin, the DSP founding member, has been studied for nearly four decades and there are still fundamental questions regarding the mode of assembly from cytosolic protein to the larger contractile machinery. All DSPs have two common domains, a GTPase or G domain, where GTP binding and hydrolysis occurs, and the stalk, comprised of the middle domain and GTPase effector domain (GED), which together drive self-assembly (Fig. 1a). Fission DSPs include an additional bundle signaling element (BSE) connecting the stalk and G domain. In dynamin, the BSE changes conformation in response to nucleotide state to drive a powerstroke that induces constriction of the helical polymer[12]. Most DSPs have additional domains to serve specialized roles within the cell. Dynamin-related protein 1 (Drp1), the master regulator of mitochondrial fission, has a unique intervening sequence adjacent to the stalk called the variable domain (VD), composed of 136 intrinsically disordered residues that confer lipid

[1]Department of Pharmacology, Case Western Reserve University School of Medicine, Cleveland, OH 44106, USA. [2]Cleveland Center for Membrane and Structural Biology, Case Western Reserve University School of Medicine, Cleveland, OH 44106, USA. [3]Department of Physiology and Biophysics, Case Western Reserve University School of Medicine, Cleveland, OH 44106, USA. [4]Center for Mitochondrial Diseases, Case Western Reserve University School of Medicine, Cleveland, OH 44106, USA. ✉e-mail: jason.mears@case.edu

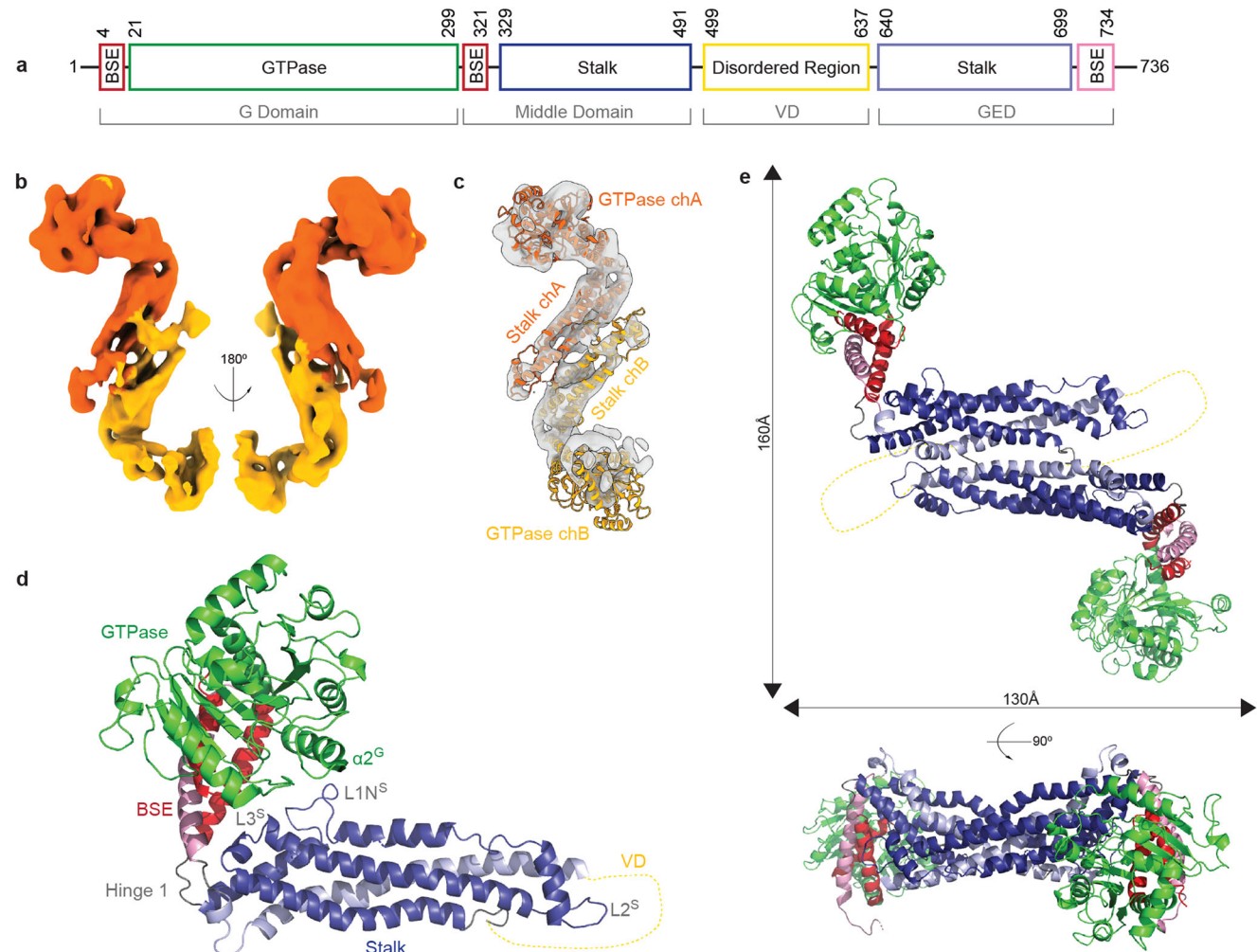

**Fig. 1 | Dimeric cryo-EM structure of Drp1. a** Previously identified domains in human Drp1 are indicated (GTPase, Bundling Signaling Element (BSE), Stalk, and Disordered Region). The corresponding residue numbers represent the architecture of Drp1 isoform 1 (Drp1-1). The DSP domains are labeled in gray. **b** Density of the dimeric cryo-EM dimer of Drp1 with a low pass filter of 6 Å applied (orange = chain A, yellow = chain B) is shown. **c** The docked chain structures within cryo-EM density after refinement are presented as ribbon diagrams. **d** Chain A (chA) of solution structure highlights the relative positions of domains in this state. The GTPase domain (green), BSE (red) and stalk region (blue) are indicated. The variable domain (VD, yellow dashed line) is not shown as this was not resolved. **e** Top down and side views of the Drp1 dimer solution structure are presented with corresponding dimensions.

sensing[13]. This region also regulates Drp1 assembly properties[14], and the disordered nature is necessary for function but likely prevents crystal formation and was deleted in the crystallographic studies[9]. This study sets out to resolve the structure of full-length Drp1 in solution. Drp1 is predominantly found in a dimer state at physiological conditions, and it is important to resolve this structure to better understand mitochondrial division[15]. Lack of structural information of a full-length DSP limits basic understanding of the regulation, inhibition, and activation of Drp1 and other DSP fission proteins.

## Results

### Identification of WT Drp1 dimer structure

Drp1 has been shown to exist as a mixture of dimers and tetramers in solution in a concentration-dependent manner, and the dimer state represents the core unit of the larger helical machinery[16]. For this study, WT Drp1 (isoform 1) was expressed in and isolated from *E. coli* using established methods[17]. This particular isoform is the second longest and includes the B insert (exons 16 and 17). It was selected for the cryoEM experiments since it has previously been shown to more dimeric when compared with other splice variants[2]. Dimers were isolated for study by diluting the protein to concentrations that would be

enriched for dimers (600 nM). Using cryo-EM, the structure of a full-length dimer of WT human Drp1 was resolved to a reported resolution of 5.97 Å and a 3DFSC resolution of 6.07 Å, which prevents the identification of side chains but secondary structure can be observed in regions of the map (Supplementary Table 1, Supplementary Fig. 1). Four conformations were identified with significant conformational heterogeneity conferred primarily through GTPase motions that highlight variability in the position of this domain relative to the stalk. Conformation 1 generated the highest quality map with the most particles, the best fit, and is used as the model dimer after applying a low pass filter of 6 Å to remove over-fitting artifacts (Fig. 1b, c). In this map both BSEs are resolved, generating the most complete model with the highest confidence. The second G domain density is not fully resolved in any of the conformations. We believe this is due to the heterogeneity of each G domain relative to one another. Conformations 2-4 provide insight in the heterogeneity in the dimer interface; however, resolution and map quality suffered due to G domain flexibility.

Overall, the architecture of the model dimer presents a compact organization of the GTPase and stalk domains (Fig. 1d, Movies 1 and 2), resulting in a ~100 Å decrease in length of the solution dimer when

compared to the length of the crystal structure dimer with the VD deletion and GPRP401-404AAAA mutation (Fig. 1e, Supplementary Fig. 1f). The compaction of Drp1 dimers, as compared to the extended state described in the crystal structure, is achieved through hinge motions. Specifically, hinge 1, comprised of two disordered loops that connect the BSE to the distal end of the stalk, adopts a conformation that places the GTPase domain near its own stalk (Fig. 1). Separately, a pivot between adjacent stalks in the homodimer increases the number of intermolecular contacts as compared to those in the crystal structure.

The largest rearrangement when comparing the solution Drp1 to previous structures was observed in hinge 1. A local resolution of 5.5 Å was observed in this region (Supplementary Fig. 1c), allowing flexible fitting of the crystal structure to the EM density. In comparison with the crystal structure, the fit revealed that the G domain is capable of significant rearrangement to bring the GTPase domain helix $\alpha2^G$ adjacent to loop $L1N^S$ in the stalk (Fig. 1d). This loop is important for Drp1 assembly, as mutations in this conserved region limit Drp1 and other DSPs ability to assemble to anything larger than a dimer[4,16,18,19]. The compaction of the GTPase domain against this loop would prevent assembly beyond a dimer, representing a conformational change compared to helical and filament assemblies that exhibit an extended G domain conformation to expose conserved self-assembly interfaces.

The residues that form the discrete dimer interface identified by the Daumke group remain at the core of this dimer conformation;[9] however, the density reveals additional burial of surface area between adjacent stalks in this region. Additional residue contacts are formed through stalk motions that close the angle between the monomers. The orientation of head and stalk is generally heterogenous and asymmetric, suggesting that the hinge motion is dynamic in the solution state of the Drp1 homodimer (Supplementary Fig. 1). The major difference in the distinct dimer conformations can be attributed to a "pivoting" motion around the fulcrum of the interface that results in altered angles between adjacent stalks. The solution dimer conformation is more closed, further obscuring potential stalk interfaces. The previous structures in the crystal lattice or assembled helical polymers likely represent an "open" conformation where the core dimer interface is maintained but the peripheral stalk regions are exposed, removing auto-inhibitory interactions that limit assembly in these regions.

## BSE lock through loop $L3^S$

Comparing the autoinhibited dimer structure to the published crystal structure, the G domain of the cryo-EM structure is positioned 79 Å closer to the distal end of the stalk. This change in position is accompanied by a 67° rotation and a 61° twist of the G domain (Fig. 2a, b). This results in a "locked" conformation of the G domain against the distal end of the stalk, mediated through interactions between the BSE and loop $L3^S$ (Fig. 2c, Movie 3). Absent side chain resolution to identify critical contacts stabilizing this conformation, mutagenesis was pursued to further examine the role of this loop. $L3^S$ was not fully resolved in the crystal structure; however, a Drp1-MiD49 filament structure[4] identified a small helical segment in the middle of the loop at residues 452-456 QELLR, and AlphaFold predicted a helix in this region as well[4,20]. A comparison of available DSP structures reveals that mitochondrial fission DSP loops have additional residues (448-451 NYST) when compared with other DSPs (Supplementary Fig. 2c–e). This additional sequence may contribute to the BSE lock, and a similar conformational rearrangement was observed by the Low group in a crystal structure tetramer of *Cyanidioschyzon merolae* Dmn1 (cmDmn1)[21], the red algae mitochondrial fission dynamin (Supplementary Fig. 2e). Dynamin has a shorter loop with a more extended $\alpha2^S$ helix, so it remains unclear whether the BSE lock is conserved in all DSPs. If this feature is unique to mitochondrial fission DSPs, this region

would affect the wide range of BSE motions depending on its activation state (Supplementary Fig. 2f).

To determine the effect of this small helical segment in $L3^S$ on the activity of Drp1, a charge reversal was introduced through a mutation, R456E, to disrupt this lock and promote the opening of the BSE away from the stalk. In a previously published structure, GMP-PCP-bound complex of Drp1 and MiD 49 reveals an extension of the G domain away from $L3^S$ (Supplementary Fig. 2f), showing the range of BSE extension possible in this region of Drp1[4]. This use of a non-hydrolysable nucleotide locks the protein in a nucleotide-bound state that promotes self-assembly state. Indeed, when GMP-PCP was added to Drp1 R456E, the protein had an increased propensity to assemble into spiral polymers compared to WT, consistent with this mutation opening the auto-inhibited lock (Fig. 2d). Negative stain images show that after 2 h of incubation, WT forms predominantly rings with short spirals averaging 0.1 μm in length. In comparison, R456E was observed to have a 10-fold decrease in the abundance of rings and a 2.6-fold increase in spiral length (Supplementary Fig. 2g–i). In agreement, 77% of the R456E protein sedimented in the presence of GMP-PCP as compared to only 52% for WT protein, further suggesting that R456E favors assembly in a GTP-bound state (Fig. 2e). The decoration on cardiolipin-containing nanotubes (CLnts) observed by negative stain was similar to WT (Fig. 2d), so the organization of the polymer does not appear to be affected by this mutation; rather, the mutation most likely results in a protein conformation that is more poised to assemble in response to nucleotide binding or lipid interactions with the VD. Importantly, there was no appreciable aggregation with the WT or R456E (Fig. 2e, f) when not in the presence of a nucleotide, and there was no shift in its solution multimeric state when assessed using size-exclusion chromatography coupled to multi-angle light scattering (SEC-MALS, Fig. 2g). Together, these data show the BSE lock did not impact the solution multimer formation. Instead, the charge reversal in $L3^S$ weakens this self-regulatory BSE lock, priming the protein for assembly. The lipid sensing regions of the variable domain and the stalk interfaces required to build the helical assembly were not perturbed.

To complement the assembly assay, enzymatic activity was measured using an endpoint malachite green assay (Fig. 2h). R456E (0.32 min⁻¹) exhibited a five-fold decrease in basal activity compared WT (1.7 min⁻¹). This change likely reflects conformational differences in the protein since the multimer state in solution is not altered. The stimulated activity of R456E in the presence of CLnts (16.8 min⁻¹) was still lower compared to WT (35.1 min⁻¹), but there was significant stimulation for the R456E when compared to its basal rate demonstrating its ability to form higher order oligomers. This is consistent with the assembly observed in the presence of GMP-PCP. Again, this mutant is more assembly-potent, and the lipid-bound structures look indistinguishable from WT (Fig. 2d), though we cannot discount small differences that affect this stimulation.

To test the effect of the mutation on mitochondrial fission in cells, we transfected Drp1 knock-out (KO) mouse embryonic fibroblasts (MEFs) with WT Drp1 and R456E (Fig. 2i). When transfected with WT Drp1, cellular mitochondrial networks were mostly fragmented due to the over-expression of the fission protein. Cells transfected with R456E were observed to have fused mitochondria, similar to the KO cells treated with an empty vector (EV) control (Fig. 2j, Supplementary Fig. 2j). Additionally, the R456E mutant protein in these transfected cells formed aggregated puncta and did not exhibit a diffuse signal observed with the WT Drp1 vector (Supplementary Fig. 2k). This likely represents premature assembly of the R456E mutant in cells, suggesting that this BSE lock is critical to sustain an auto-inhibited state that prevents premature assembly of Drp1 polymers. Only when this autoinhibited state is relieved, or unlocked, does the WT protein become primed to form a helical assembly around the outer mitochondrial membrane.

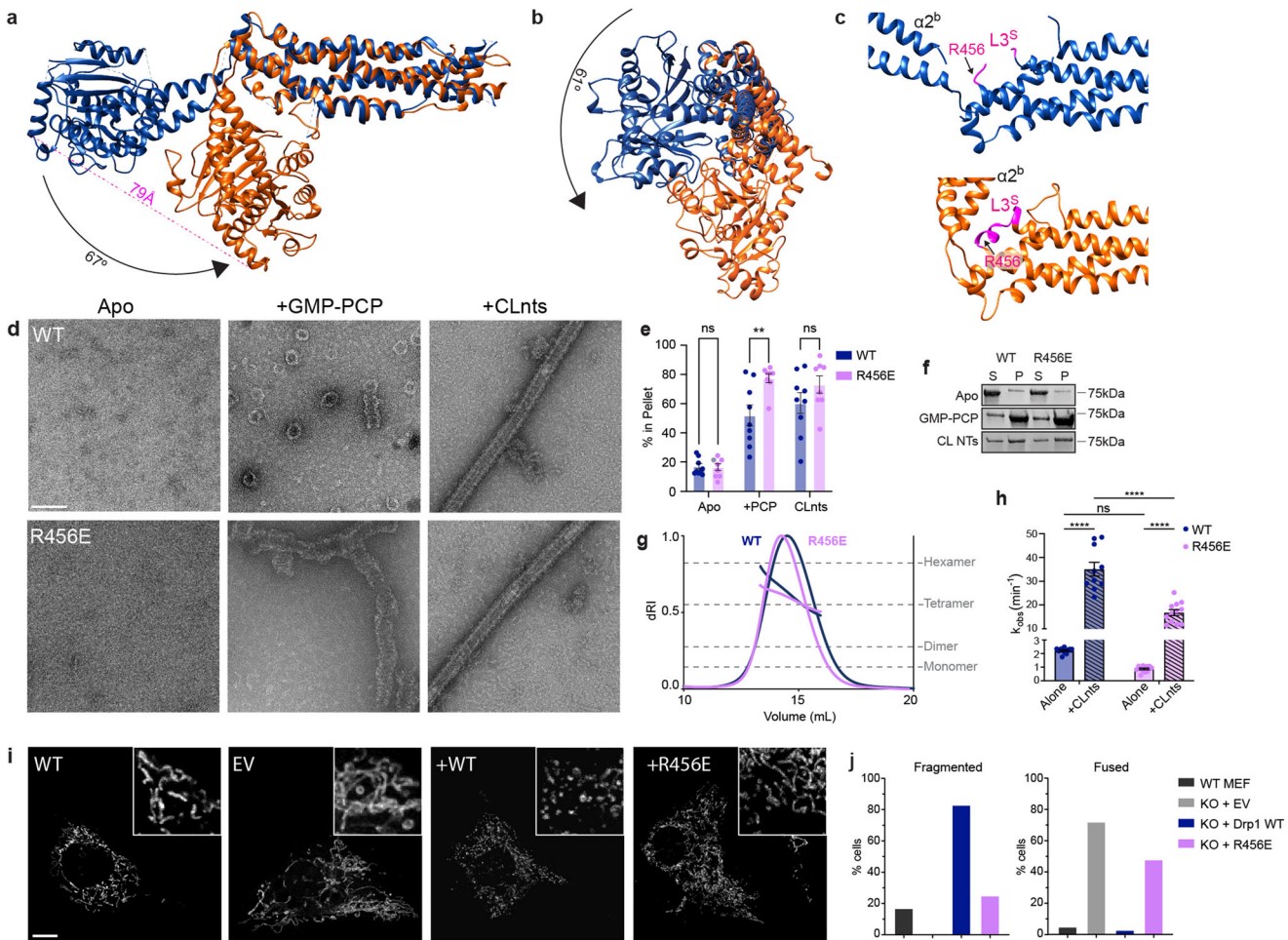

**Fig. 2 | The BSE lock. a, b** Comparing the previous crystal structure[9] (PBD ID: 4BEJ, blue) to the dimeric cryo-EM structure (orange) based on overlay of the stalk region. **c** Loop 3 (L3$^s$) interacts with BSE helix α2$^b$ and contains R456. **d** Negative stain electron microscopy characterization of WT and R456E Drp1 in the presence and absence of non-hydrolysable GMP-PCP and CL-containing nanotubes (CLnts). Multiple grids were made and several images were collected for each condition (WT apo=8, WT + GMP-PCP = 38, WT+CLNTs = 44, R456E apo = 4, R456E + GMP-PCP = 21, R456E+CLNTs=48). Scale bar = 100 nm. **e, f** Sedimentation assays were used to quantify changes in polymerization based on the relative percent of protein detected in the supernatant (S) and pellet (P) fractions. Data are presented as mean values +/− SEM, and a two-tail *t*-test was used to determine statistical significance. Each dot represents an experimental replicate (WT(blue) = 9, R456E(pink) = 8. ** *P* = 0.006). **g** SEC-MALS analysis of Drp1 WT (blue) and Drp1

R456E (pink) is presented with multimeric states indicated. **h** GTPase activity was determined for WT (blue) and Drp1 R456E (pink) alone and stimulated with CLnts. Data are presented as mean values +/− SEM, and a two-tail *t*-test was used to determine statistical significance. Each dot represents an experimental replicate (WT (apo)=12, WT (CLnts)=10, R456E (apo and CLnts)=12. **** *P* < 0.0001).
**i, j** MitoTracker Orange was used assess mitochondrial morphology in WT MEFs compared to Drp1 knock-out MEFs transfected with an empty pCMV vector (EV) and pCMV vectors containing Drp1 WT and Drp1 R456E. Scale bar 5 μm, inset 10 × 10 μm. Mitochondrial morphology was quantified using a blinded assessment described in the methods section. Two independent experiments were performed to assess the percentage of cells with defined mitochondrial morphologies in each sample (Total cells counted: WT MEF (dark gray) = 60, EV (light gray) = 64, KO+Drp1 (WT, blue) = 30, KO + R456E (pink) = 33).

## A flexible dimer interface

The dimer interface that orients the monomers relative to one another was found to have a large amount of heterogeneity within the conformations and among available structural data. The crystal structure exhibits a discrete dimer interface with a more acute lateral angle measuring 85° (Fig. 3a). The solution structure presented here suggests that removing the VD and introducing the poly-A mutation within L2$^S$ near the membrane proximal interface (previously labeled interface 3) yielded an open conformation more amenable to crystallization. These changes also disrupted key regulatory regions, leading to an "open" conformation and the stalk orientations are consistent with previously reported DSP helical conformations. Within the cell, interactions with lipids, ER contact sites, post-translational modification(s), and partner proteins interactions could all promote an open state alone or in concert with one another. In the solution (i.e., "closed") state, L2$^S$ is juxtaposed to α1N$^s$, and this interaction requires a conformation with an obtuse angle between adjacent stalks (ranging

from 103 to 145°) to form a more continuous interface (Supplementary Fig. 3a). In order to identify intra-monomer conformational rearrangements, the four helices comprising the stalk of the solution structure were aligned with the crystal structure helices (Fig. 3b). Loops within α1 confer a large degree of flexibility, and this is evident when comparing the different chains of the crystal structure (Supplementary Fig. 3b). Helices 2 and 4 were relatively unchanged, and Helix 3 had a shift in alignment at the C-terminal end due to flexion in the middle of the helix. A conserved tyrosine (Y493) was found to bookend the dimer interface based on optimal placement of this helix in the density (Fig. 3c, d, Movie 4). To alter the interactions at this position, mutagenesis substituted a smaller alanine in place of the bulkier tyrosine. As evidenced by negative-stain EM analysis, this mutation showed a decreased ability to form spirals in the presence of GMP-PCP and was unable to uniformly decorate CLnts when compared to WT (Fig. 3e). This finding was further quantified using a sedimentation assay. In apo conditions, both WT and Y493A resulted in similar levels of protein

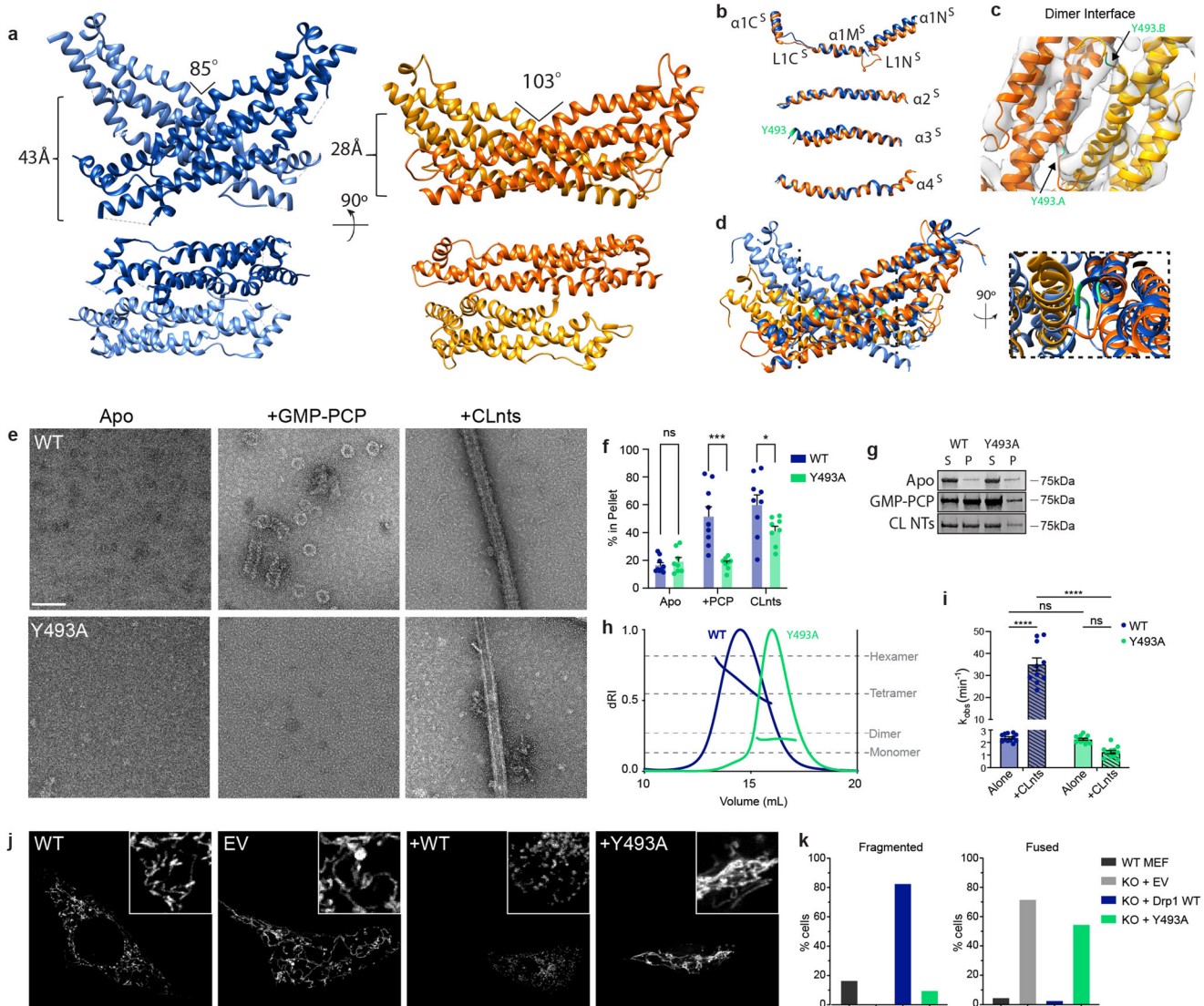

**Fig. 3 | Dimeric cryo-EM structure dimer interface. a** Comparison of the stalk orientations in the crystal structure dimer interface, (PDB code: 4BEJ, dark blue - chain A, light blue - chain B) and the dimeric cryo-EM structure (dark orange - chain A, light orange - chain B). **b** Alignment of each helix comprising the stalk domain (PDB code: 4BEJ, blue; cryo-EM structure, orange). Y493 is highlighted in light green. **c** Helices forming the dimer interface are fit within the cryo-EM density (chain A, orange; chain B, yellow). Y493 (green) bookends the interface. **d** Alignment of crystal structure dimer and dimeric cryo-EM structure stalks, reference chain is chain A. Zoomed in view, Y493 in light green. **e** Negative stain electron microscopy characterization of WT and Y493A Drp1 in the presence and absence of non-hydrolysable GMP-PCP and CL-containing nanotubes (CLnts). Multiple grids were made and several images were collected for each condition (WT apo=8, WT + GMP-PCP = 38, WT+CLNTs = 44, Y493A apo = 9, Y493A + GMP-PCP = 25, Y493A+CLNTs=46). Scale bar = 100 nm. **f, g** Sedimentation assays were used to quantify changes in polymerization based on the relative percent of protein detected in the supernatant (S) and pellet (P) fractions. Data are presented as mean values +/− SEM, and a two-tail $t$-test was used to determine statistical significance. Each dot represents an experimental replicate (WT (blue) = 9, Y493A (green) = 8.***$P$ = 0.0005; *$P$ = 0.04). **h** SEC-MALS analysis of Drp1 WT (blue) and Drp1 Y493A (green) is presented with multimeric states indicated. **i** GTPase activity was assessed for WT (blue) and Drp1 Y493A (green) alone and stimulated with CLnts. Data are presented as mean values +/− SEM, and a two-tail $t$-test was used to determine statistical significance. Each dot represents an experimental replicate (WT (apo) = 12, WT (nts) = 10, Y493A (apo) = 12, Y493A (nts)=11. ****$P$ < 0.0001). **j, k** MitoTracker Orange was used assess mitochondrial morphology in WT MEFs compared to Drp1 knock-out MEFs transfected with an empty pCMV vector (EV) and pCMV vectors containing Drp1 WT and Drp1 Y493A. Scale bar 5 μm. Inset 10 μm. Mitochondrial morphology was quantified using a blinded assessment described in the methods section. Two independent experiments were performed to assess the percentage of cells with defined mitochondrial morphologies in each sample (Total cells counted: WT MEF (dark gray) = 60, EV (light grey) = 64, KO+Drp1 WT (blue) = 30, KO + Y493A (green) = 29).

found in the pellet (17% and 20% respectively). No assembly of Y493A was detected when GMP-PCP was added to induce spiral formation, as only 19% of the Y493A sediments with GMP-PCP. This represents no change compared to average sedimentation under apo conditions, while WT Drp1 had a three-fold increase in pelleted protein (52%). A decrease was also observed in the ability of Y493A to decorate CLnts (42% of Y493A was found in the pellet compared to 60% for WT), confirming that this mutant prevents assembly even though lipid

binding is likely preserved since the VD is unchanged. This is not surprising, considering the importance of the dimer state, the functional unit of Drp1. If a stable dimer interface is disrupted, the protein will be incapable of forming larger assemblies. Since no significant change in sedimentation was observed when comparing the WT and Y493A Drp1 in the absence of assembly inducers (Fig. 3f, g), SEC-MALS was used to assess the multimer state in solution. Y493A was found to exist in equilibrium between dimers and monomers and was not able to form

larger multimers observed with WT Drp1 at the same concentration (Fig. 3h). Knowing that the multimer state for Drp1 is concentration dependent, mass photometry was used at a lower concentration (100 nM) to confirm this difference, and Y493A was found to exist largely as a monomer while WT protein was mostly dimeric (Supplementary Fig. 3c).

The GTPase activity was assessed for Y493A and WT Drp1 (Fig. 3i), and the basal rates in solution were comparable (2.3 min$^{-1}$ for Y493A versus 2.4 min$^{-1}$); however, CLnts stimulation was only observed with the WT protein (35 min$^{-1}$), while Y493A was not stimulated (1.2 min$^{-1}$). Therefore, Y493A maintained basal GTPase activity but was incapable of functional assembly.

Transfecting Drp1 KO MEF cells with Y493A saw no change in the interconnected, fused mitochondrial morphology when compared to the empty vector control. Conversely, WT Drp1 transfection resulted in a fragmented mitochondrial network (Fig. 3j, k, Supplementary Fig. 3d). This observation is consistent with the destabilization of the continuous dimer interface by the Y493A mutation that was introduced, which prevents functional assembly and limits mitochondrial fission in cells. Therefore, the Y493 residue is critical for stabilizing interface 2, bookending either side of the interface to accommodate flexibility from a continuous interface to a more discreet intermolecular interaction required for spiral and helical oligomerization.

## Discussion

This cryo-EM solution structure of WT Drp1 demonstrates a previously unappreciated inactive dimer conformation through additional self-regulatory interactions. With other DSPs, conformational changes were observed in the BSE hinge in response to nucleotide binding and partner protein interactions[4,12]. These motions were proposed to alter stalk interactions to mediate constriction of the helical lattice. The solution structures of Drp1 demonstrate additional crosstalk between the G domain and stalk regions that offer a role for accessory protein and lipid interactions in relieving autoinhibitory conformations (Fig. 4). DSPs form helical assemblies through two highly conserved interfaces, interface 1 and interface 3 (Supplementary Fig. 4). Using dynamin 3 interfaces as a reference, we aligned two dimers of the Drp1 crystal structure and two dimers of the cryoEM structure to interface 1 (Fig. 4a, b). The crystal structure tetramer aligned with no apparent clashes. Interface 3 was also found to be in a reasonable position relative to interface 1. When the cryoEM dimers were aligned to the dynamin 3 interface 1, it becomes apparent that the G domains are in a vastly different conformation compared to the crystal structure. Additionally, interface 3 is not aligned when interface 1 was the reference. To determine the possible clashes, interfaces 1 and 3 were aligned separately. In both alignments, obvious steric clashes were observed. Helices α1C$^S$ and α1N$^S$ and loops L1N$^S$ and L2$^S$ would prevent the dimeric cryoEM structure from forming either interface without undergoing conformational rearrangements. This is informative as mutations found in these two loops have been characterized to limit assembly.

Based on our structural and biochemical data, the BSE lock remains largely engaged to prevent premature assembly in the absence of an activating stimulus. This conformation is associated with increased stalk interactions, forming a more continuous dimer interface. Together, these changes compared to the crystal structure represent key regulatory motions that open the dimer preceding functional assembly. These motions leading to an "open" state are likely stabilized by nucleotide and/or partner protein interactions that promote assembly of the fission machinery.

To this point, a nucleotide-bound, cryo-EM structure of a Drp1-MiD49 co-polymer was solved with full-length Drp1 in an open conformation[4], and it also exhibited a more extended BSE due to partner protein binding interactions at stalk interfaces (Supplementary Fig. 3g). When comparing the solution structure with the MiD49-bound filament, the "closed" BSE lock is incompatible with two MiD49 interfaces (Supplementary Fig. 3h). One proposed MiD49 interface is positioned at loop 1N$^S$, which would require that the G Domain extend to avoid steric clashes, consistent with altered hydrolysis, and be available for G-G interactions that stabilized inter-rung interactions in the helical lattice. Since MiD49 is anchored at the surface of mitochondria, this interaction would favor opening of Drp1 to promote assembly at these defined membrane sites.

Nucleotide interactions could also facilitate opening of hinge 1 through motions in the adjacent BSE hinge, and this model is consistent with nucleotide-induced formation of spiral structures observed when Drp1 is incubated with non-hydrolysable GTP analogs. It remains unclear whether the BSE hinge responds to nucleotide binding, since nucleotide binding does not lead to constriction like it does for dynamin. Rather, nucleotide binding likely augments release of the BSE lock to promote assembly. Thus, protein or nucleotide interactions change the G Domain conformation to inform conformational rearrangements in the stalk that permit assembly of the helical fission complex.

A separate MiD49 interface at loop 2$^S$ is likely regulated by interactions with the VD. In the Drp1-MiD 49 structure, the binding of the partner protein would occlude VD interactions as previously suggested[13]. In the solution structure, L2$^S$ interactions stabilize a more continuous dimer interface and juxtapose the VDs from each monomer on opposite ends of the dimer structure, placing the disordered regions far away from one another. Deletion of the VD in the crystal structure removed this large regulatory domain, and previous studies have shown that removing the VD predisposes Drp1 to self-assembly through interactions with partner proteins[22] or when the protein concentration is elevated[23]. On the surface of mitochondria, VD engagement with the membrane would promote opening of the dimer interface and expose key intermolecular interaction sites to drive helical formation.

In this model for assembly of the mitochondrial fission machinery, locked G domains extend to promote helical assembly. In addition, the more continuous dimer interface observed in solution exhibits a wide range of flexibility within the dimer. This conformation occludes residues necessary for helical assembly, decreasing the allowable geometries needed to build the fission machinery. We propose that these self-regulatory interactions exist in an equilibrium in solution, sampling various conformations. Assembly-limited mutants, often located in the L1$^N$ loop, likely limit the range of conformational sampling and thereby prevent opening of the dimer to an active state. Conversely, stimulating interactions with specific lipid and partner proteins at the surface of mitochondria open the stalk configuration to promote intermolecular contacts between Drp1 dimers, which have been shown to be the core building block for the contractile helical polymer. Post-translational modifications could bias conformational sampling in the solution dimer to augment or prevent functional assembly. In this way, partner protein and lipid binding affinities reflect the accessibility of interaction sites on the Drp1 stalk that dictate further opening of the Drp1 dimer to build the mitochondrial fission apparatus in a manner that is spatiotemporally regulated. These insights will be important for developing new selective inhibitors of Drp1. To date, efforts to regulate Drp1 have been based on the open and active conformation of Drp1 and have failed to produce a reliable inhibitor targeting Drp1 directly. In the future, factors that stabilize regulatory interactions in the dimer state would limit assembly of the fission machinery and provide a novel approach to prevent mitochondrial fission.

## Methods

### Protein constructs and mutagenesis

Drp1 Isoform 1 (UniProt ID O00429-1) was cloned into the pCal-N-EK vector as described previously[17,22]. Site-directed mutagenesis for Drp1

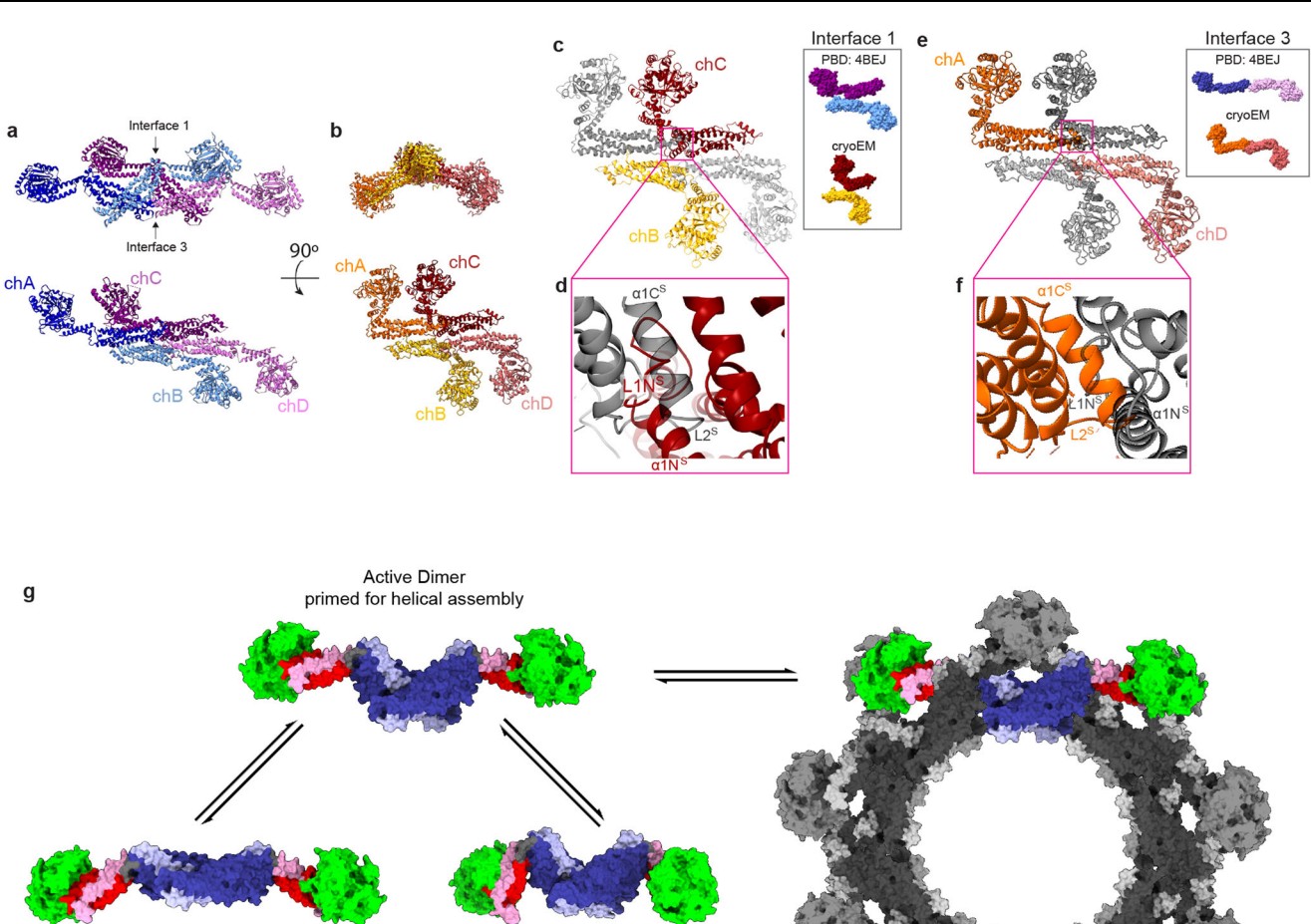

**Fig. 4 | Model for Drp1 regulation. a** Two dimers from the crystal structure (PDB ID: 4BEJ) are aligned to the dynamin interface 1. **b** Two dimers of the dimeric cryoEM structure are aligned to the dynamin interface 1. **c** Top down view of the dimeric cryoEM structure aligned to the dynamin interface 1. Chains not involved in the interface are in gray. Inset box provided for reference, crystal structure interface 1 top, dimeric cryoEM structure interface 1 bottom. **d** Steric clashes between adjacent chains in the cryoEM structure would prevent proper interface alignment of interface 1. **e** Top-down view of the dimeric cryoEM structure aligned to the dynamin interface 3. Chains not involved in the interface are in gray. Inset box provided for reference, crystal structure interface 3 top, dimeric cryoEM structure interface 3 bottom. **f** Steric clashes between adjacent chains in the cryoEM structure would prevent proper interface alignment of interface 3. **g** The dimeric cryoEM structure (bottom left) is in an autoinhibited state. The BSE lock and the closed dimer interface are maintained by weak intramolecular interactions. Intermolecular interactions between Drp1 and partner proteins, organelle contact sites, and/or lipid interactions relieve inhibition to promote an assembly-primed conformation. Helical assembly requires both an open dimer interface and an unlocked, extended G Domain. GTPase domain in green, stalk in dark blue, GED in light blue, BSE in red (N-terminal), and Pink (C-terminal). Variable domain not shown.

R456E and Y493A was completed to introduce each mutation into this construct individually using the QuikChange Lightning kit (Agilent) with primers from Integrated DNA Technologies (Coralville, IA).

## Protein expression and purification

All Drp1 constructs were expressed in BL21-(DE3) Star *Escherichia coli*. Cells were grown in LB containing 100 µg/mL ampicillin at 18 °C with shaking at 200 rpm for 24 h after induction with 1 mM isopropyl-1-thio-β-D-galactopyranoside (IPTG). Then, cells were harvested via centrifugation at 4300 × g for 20 min at 4 °C. The resulting pellet was resuspended in CalA Buffer (0.5 M L-Arginine pH 7.4, 0.3 M NaCl, 5 mM MgCl₂, 2 mM CaCl₂, 1 mM imidazole, 10 mM β-mercaptoethanol) with 1 mM Pefabloc-SC and 100 µg/mL lysozyme. Cells were lysed by sonication on ice. Next, the cell debris was pelleted via centrifugation at 150,000 × g for 1 h at 4 °C. First, the CBP-tagged Drp1 was purified by affinity chromatography using calmodulin agarose resin (Agilent) that had been pre-equilibrated with CalA Buffer. After the supernatant was loaded onto the column, the resin was washed with 25 column volumes of CalA Buffer. Next, 8 fractions of eluent were collected using 0.5 column volumes of CalB Buffer (0.5 M L-Arginine pH 7.4, 0.3 M NaCl, 2.5 mM EGTA, 10 mM β-mercaptoethanol). Protein-containing fractions were pooled and incubated with GST-tagged PreScission

Protease (HRV-3C) overnight at 4 °C to remove the CBP-tag. This solution was concentrated using a 30,000 molecular weight cut-off centrifugal filter (Amicon). This concentrated pool of Drp1 was further purified by size exclusion chromatography (SEC) with an ÄKTA Purifier FPLC (GE Healthcare) and a HiLoad 16/600 Superdex 200 Prep Grade column that had been pre-equilibrated with SEC Buffer (25 mM HEPES (KOH) pH 7.5, 0.15 M KCl, 5 mM MgCl$_2$, 10 mM β-mercaptoethanol). All elution fractions containing Drp1 were pooled and concentrated once again, and glycerol (5% final) was added. The purified Drp1 was aliquoted, flash frozen in liquid nitrogen, and stored at −80 °C until use.

## Cryo-EM sample preparation, imaging, and processing
Holey carbon grids (1.2/1.3, quantifoil) were coated with 0.2 mg/ml graphene oxide (GO) support film prepared in house. Drp1 was diluted to 0.05 mg/ml, incubated on the GO grid for 30 s, and then frozen using a FEI Vitrobot Mark IV with a blot force of 0 and a blot time of 2.0 s. Grids were imaged on a Titan Krios (300 keV) equipped with a K2 Summit camera at a magnification of 130,000x. Data were collected over 40 frames for 8 s at a nominal dose of 49 e/Å$^2$ (6.11 e/p/s).

All data was processed using cryoSPARC, version 3[24]. Movies were motion corrected using MotionCor2[25], and Patch CTF was used for contrast transfer function (CTF) estimation. Initial particles were picked manually (2054) and then used to train TOPAZ[26]. Initial processing selected a subset of junk particles and good particles. Those were used to create to reference densities. All particles were then sorted using these references. 3D refinement was completed using the Nonuniform refinement job in cryoSPARC[27]. Local resolution was determined using the Local Resolution job in cryoSPARC and visualized with Chimera[28].

## Model building, flexible fit, and refinement
The starting model was built using the AlphaFold predicted monomer[20,29]. Two monomer chains were aligned to the crystal structure's dimer interface (4BEJ)[9]. This dimer model was docked using rigid body docking in Chimera. Then in VMD, an all-atom model was generated using the Automatic PSF Builder in VMD[30]. Files were prepared for NAMD processing following previously described methods[31]. The four density maps were converted to an MDFF potential. Secondary restraints were applied using NAMD's extrabonds feature and plugins to preserve secondary structure and to limit artifacts to chiral centers and cis peptide bonds. The simulations used a GScale was set to 0.3, temperature was set to 300, and the number of timesteps (numsteps) was 1,000,000–1,500,000. A minimization step was run with a GScale was set to 10, temperature was set to 300, and minimize steps (minsteps) was set to 2000. The model was then minimized and refined using Phenix Real-space refinement[32,33] to fix clashes and outliers.

## Negative-stain electron microscopy
All samples were added to 400 mesh copper grids with a formvar carbon film (EMS FCF400-CU) and stained using 2% uranyl acetate. Each grid was made using 2 μM protein and either 1 mM GMP-PCP or 150 μM lipid nanotubes as indicated and were left to incubate for at least 60 min at room temperature. Sample images were acquired on a Tecnai TF20 electron microscope (FEI Co.) at 200 keV, respectively. The TF20 was equipped with TVIPS F-416 CMOS (4k × 4k) camera and images were acquired at a magnification of 30,000×.

## Malachite green colorimetric assay
The basal GTPase activity of Drp1 was measured using a colorimetric assay to detect released phosphate, as described previously[17,22]. Briefly, Drp1 (500 nM final) was diluted to 2.4X with Assembly Buffer (25 mM HEPES (KOH) pH 7.5, 150 mM KCl, 10 mM β-mercaptoethanol). To start the reaction, 3X GTP/MgCl$_2$ (1 mM and 2 mM final, respectively) was added to the Drp1 with either 4X lipid (150 μM final)

to calculate the lipid-stimulated rates or only Assembly Buffer to calculate the rate for the protein alone in solution. The reaction was carried out at 37 °C. At the chosen time points, a sample aliquot was taken and quickly added to EDTA (100 mM final) to stop the reaction. The time points used to calculate the rate for the protein alone in solution were 5, 10, 20, 40, and 60 min. For the lipid-stimulated rates the time points were 2, 4, 6, 8, and 10 min for all samples except Y493A + CLnts, which was instead measured using the longer 5, 10, 20, 40, 60 min time course. After collecting all time points, Malachite Green Reagent (1 mM malachite green carbinol, 10 mM ammonium molybdate tetrahydrate, 1 N HCl) was added to each sample, and the absorbance at 650 nm was measured using a VersaMax microplate reader (Molecular Devices).

## Lipid nanotube synthesis
All lipid nanotubes utilized here were comprised by 40% D-galactosyl-beta-1′-N-nervonyl-erythro-sphingosine (GC), 35% phosphatidylethanolamine (PE). 25% bovine heart cardiolipin (CL) molar fractions. All lipids were purchased from Avanti Polar Lipids, Inc. (Alabaster, AL). Lipids were added to a glass test tube and slowly dried to a thin film using nitrogen gas. The lipid film was then stored in a desiccator for at least 1 h to ensure any trace solvent remaining was removed. Then the lipid film was rehydrated with a buffer (200 μL) containing 50 mM HEPES (KOH) pH 7.5 and 0.15 M KCl and heated in a 37 °C water bath for ~40 min with gentle vortexing every 10 min. With these volumes, the final lipid nanotube concentration was 2 mM. The lipid film was placed in a water bath sonicator for 30 s and the resulting nanotubes were stored on ice until use.

## Size exclusion chromatography with multi-angle light scattering (SEC-MALS)
SEC-MALS experiments were performed as before[22]. Briefly, 5 μM Drp1 was injected onto a Superose 6 10/300GL column in an ÄKTApure FPLC system (GE Healthcare) connected in line with DAWN Heleos-II 18-angle MALS and Optilab T-rEX differential refractive index (dRI) detectors from Wyatt Technology. Data were analyzed with ASTRA 7 software from Wyatt Technology.

## Sedimentation assay
To quantify Drp1 oligomerization, a sedimentation assay was conducted similar to what has been described previously[34,35]. Large oligomers formed by Drp1 samples, in the presence of ligands, were found in the pellet after a medium speed centrifugation. Specifically, protein was diluted in HEPES KCl buffer to 2 μM, and specified WT and mutant samples were incubated at room temperature with lipid nanotubes (150 μM) and/or GMPPCP (2 mM) for at least 60 min. The mixtures were then spun at 15,000 rpm for 10 min in a tabletop centrifuge (Eppendorf). The supernatant and pellet fractions were separated, collected, and immediately mixed with SDS-PAGE loading dye (Bio-Rad) and heated briefly at 100 °C. These samples were run on an SDS-PAGE gel and stained with an InstantBlue Coomassie dye (Expedeon). Gels were scanned using an Odyssey XF Imaging System (Li-Cor) and densitometry analysis was done using the Image Studio Lite Ver 5.2.

## Cell lines and transfection protocol
WT and Drp1 KO Murine embryonic fibroblasts (MEFs)[36] were cultured in complete media (DMEM with 10% FBS supplemented with Penicillin/Streptomycin, L-Glutamine, Insulin-Transferrin-Sodium Selenite Supplement (ITS), αFibroblast Growth Factor (FGF), and Uridine). Drp1 constructs were cloned into a pCMV-Myc vector. 25,000–50,000 cells were plated and transfection of specified constructs or empty vector (6–13 μg/nL) was performed with Lipofectamine 2000 (Thermo Fisher Scientific, 1166819) and Opti-MEM (Gipco, 31985) for 24 h.

## Immunofluorescent staining and assessing mitochondrial morphology

Cells were plated on glass-bottom dishes and were stained with 250 nM of MitoTracker™ Orange CMTMRos (Invitrogen, M7510) for 30 min then fixed in 4% paraformaldehyde (Alfa Aesar, 43368) for 15 min and stained with DAPI in PBS 1:10,000 (Invitrogen, D1306) for 5 min. 3 washes were performed between each step to reduce background. Cells were permeabilized with 1% triton, blocked with 10% BSA, and incubated overnight with 1:250 c-Myc antibody (Santa Cruz, sc40) in 10% BSA. They were then washed with PBS 3 times, 5 min each wash and treated with 1:500 Alexa Fluor Plus 488, highly cross absorbed (Invitrogen, A32723TR) in 10% BSA. Then washed 3 times with PBS. Images acquired using a Leica SP8 Gated STED confocal Microscope in the Light Microscopy Imaging Facility at Case Western Reserve University. Blinded individuals independently categorized each cell as either fused, tubular, intermediate, or fragmented for randomized confocal images.

## Statistics

Statistics were done in Prism GraphPad using a two-tail *t*-test. All measurements were taken from distinct samples as biological replicates.

## Reporting summary

Further information on research design is available in the Nature Portfolio Reporting Summary linked to this article.

## Data availability

The data that support this study are available from the corresponding authors upon request. The cryo-EM map has been deposited in the Electron Microscopy Data Bank (EMDB) under accession codes EMDB-40967 (Drp1 solution dimer). The atomic coordinates have been deposited in the Protein Data Bank (PDB) under accession codes 8T1H (Atomic model for Drp1 solution dimer). Other structures referenced in this manuscript are indicated throughout and include PDB ID: 4BEJ, 5WP9, 6FGZ, 6DLV, 3SZR, and 5A3F. A Source Data file is available with this manuscript. Source data are provided with this paper.

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

## Acknowledgements
We would like to acknowledge use of the (Leica SP8 confocal microscope) in the Light Microscopy Imaging Facility at Case Western Reserve University made available through the Office of Research Infrastructure (NIH-ORIP) Shared Instrumentation Grant S10OD016164. We would like to acknowledge the West/Midwest Consortium for High-Resolution Cryo Electron Microscopy at the University of California, Los Angeles for use of their facilities for data collection made available through the cooperative agreements grant (NIH U24) GM116792. K.R. discloses support for the research of this work from the National Institute of General Medical Sciences [F31 GM139324]. J.A.M. discloses support for the research and publication of this work from National Institute of General Medical Sciences [R01 GM125844]. R.R. discloses support for the research and publication of this work from National Institute of General Medical Sciences [R01 GM121583]. E.Y. discloses support for the research and publication of this work from National Institute of Allergy and Infectious Diseases [R01 AI145069]. Molecular graphics images were produced using the UCSF Chimera package from the Resource for Biocomputing, Visualization, and Informatics at the University of California, San Francisco (supported by NIH P41 RR-01081). We would like to thank Harry Scott for his expertise in sample preparation and screening.

## Author contributions
K.R. and J.A.M. conceived and planned the experiments. K.R., B.L.B., N.A.R., Y.B., R.R., and M.S.K.S. carried out the experiments. C.S. and W.H. assisted with data processing and model validation. K.R. designed the figures and drafted the manuscript with supervision of J.A.M. and input from all authors. R.R., E.W.Y., and D.J.T. contributed to the interpretation of results.

## Competing interests
The authors declare no competing interests.
