## [Peer Review File · Nature Communications]

Structural Basis for Regulated Assembly of the Mitochondrial Fission GTPase Drp1Reviewer #1 (Remarks to the Author):

In this study, Rochon et. al. present a cryo EM structure of full length Drp1 in a unique dimeric conformation, which is the authors interpret as representing an autoinhibited conformation since two previously established oligomerization interfaces are sterically blocked in this dimeric conformation. They identify two key conformational features that seem critical for this interpretation: the BSE lock and stalk interface. According to the introduction, it remained largely unknown how Drp1 and other dynamin proteins regulate their assembly into higher order oligomers when stimulating mitochondrial fission. This study attempts to fill this gap in knowledge by providing a view of an autoinhibited conformation. This conformation differs from previous crystal structures, and this new viewpoint may be of value to the dynamin field. Mutations at both sites were introduced to disrupt this autoinhibited conformation, and assessed in numerous ways including assembly in negative stain EM, enzymatic GAP assays, and transfection/microscopy studies. Taken together, these provide an experimentally comprehensive approach to the question, and only a few questions are raised below regarding the experimental data.

The studies seem generally technically sound, though I cannot comment directly on the details of the cryo EM due to limited expertise in this technique. The structural analysis, while performed appropriately, are unfortunately not consistently presented in a clear way that is easy to follow; I often found myself rereading sections in an attempt to grasp the intended analysis. I do believe that a careful reworking of some sections of the text, and/or some new figures or more clearly labeled panels, may allow this paper to appeal to a broader audience beyond just the specifics of dynamin structure; without this, I do not believe this study is quite ready for publication.

The conclusions also seem overly drawn, with alternative explanations not always provided. While the provided data do not argue against the mechanism presented, this interpretation is not the only possible explanation. For example, the instances of "primed for assembly" versus observable assembly versus oligomerization were difficult to follow.

Additionally, other than two GTPase assays, I do not believe that a strong connection between GTPase activity, GTPase domain conformation, and autoinhibitory/active conformational differences are explored enough. Does the G domain have the canonical switch regions, and are these observed in the cryo EM density? Is the interaction with the BSE lock dependent on the switch regions?

Lastly, the abstract should be edited to include the findings beyond just the cryo EM structure.

SPECIFIC QUESTIONS/EDITS

1. Line 46. Can the authors please provide one or two additional sentences to frame the study; for example, to the current gap in the knowledge specifically for Drp1 and how the study directly addresses this?
2. If 4 conformations were obtained, which one is the predominant structure being used for analysis? For example, Figure 1b. Some explanation of how this was chosen would be appropriate. Also, is the BSE lock discussed later on consistent in all these conformations?
3. Lines 59-89. This whole section would benefit from more illustration. For example, how is larger assembly sterically blocked by the observed dimer conformation? Specifically, what is occluding the DSP helical structures that would otherwise mediate higher assembly; is is the orientation of the GTPase domain? Is this true for all 4 conformations? This is a main conclusion of the study and yet it is does not seem to be illustrated clearly.
4. Lines 61-66 discussion of hinge motion: Figure panels illustrating this should be explicitly stated in text. Similar problem with reading lines 77-89. Illustrations are needed.
5. Figure 1d it appears as though the GTPase density in chain B is weak/missing. Please add comments and interpretations of this in the text.
6. Line 91, please explain which structures are being compared when saying "The G domain moves". From the Figure 2 legend, I see it is the crystal structure vs cryo EM structure, but what is the

Alphafold component being cited in the figure legend?

7. Line 115-, please explain the use of GMP-PCP in the experiment. Is the result meant to imply that the GTP-bound conformation of the G domain releases the autoinhibited state? Did you confirm that Drp1 in this sample is bound to GMP-PCP? Related, in the methods, please describe how exchange to GMP-PCP was performed and detected. Does the Drp1 purify with GDP bound?

8. Was cryo EM attempted on the R456E GMP-PCP protein to compare with the previous study of ref 4?

9. Fig 2d, 2 μ M protein was used for negative stain grid. This is above the aforementioned 600 nM concentration used to enrich for dimers (line 51). Can you comment on the dimer versus tetramer in this sample, and whether this affects the results of the WT versus mutant?

10. Fig 2e. All data points should be shown overlaying bar graph. I am not convinced by the bar charts and error bars that the mutation is statistically different than WT in the PCP samples but not the CLnts. Also, the conclusion drawn is that release of the BSE lock is weakened with mutation, but that multimer formation is not affected; can this result be reconciled with the overall conclusion that the observed dimer is in the autoinhibited conformation? What is the difference between multimerization (not enhanced) versus assembly (which is "primed" in the mutant protein sample)?

11. Line 131-140. I am not sure that I agree with the interpretation that the fold increase of the mutant is functionally different than the WT. The mutant is still less active than the WT; is the mutant protein less stable? A small point: to aid in reading, please include in the text that the enzymatic activity being measured is the GTPase activity of the G domain using an endpoint malachite green assay.

12. Lines 141-151 and Figure 2i-j. I have concerns with the methodology described regarding the blinded individuals scoring the mitochondrial phenotypes. If four categories were used, why are only two shown? How indistinguishable are the phenotypes from each other? Did multiple blinded individuals score the same datasets, and if so, what were the discrepancies across their scoring? The dataset seems quite small (N = 20s to 40s only); was this N for each individual transfected cell, or transfected wells, or experiments? Is there a way to automatically score this phenotype? The bar graph reflects that the WT cells are neither fragmented nor fused; which category do the majority of WT MEF's belong to? CTCF is shown, but was this applied to the cells being scored?

13. Extended Figure 2h needs labels. The WT cell (presumably on the left panel) looks fuzzy rather than diffuse. Extended Figure 2i, is the expression of R456E mutant higher? What might this mean for the observed phenotypes? Also, an anti-myc immunoblot should be performed to complement this CTCF and also confirm that the expressed proteins are full-length. Together with point 12 above, I feel the data are over interpreted with respect to the autoinhibited conformation hypothesis.

14. Was a mutant in the corresponding surface on the alpha 2b considered?

15. The description of the dimer interface would benefit from more illustrations and/or simpler text. As it stands, the text is confusing with respect to the heterogeneity and comparison with the previous crystal structures. The main point of this section, that Y493 is at the interface, is supported by weak density in Extended data Figure 3b; this panel should nonetheless be moved to the main figure due to its importance for setting up the remaining experiments.

16. I was surprised a modest Tyr to Ala mutation was enough to elicit this effect; what is known about the affinity of dimerization?

17. Line 201: there is little evidence for the "shock absorber" mechanism. Can this be explained more clearly?

MINOR Comments

1. Line 46, please provide references for crystallographic studies where the VD loop was removed.

2. Lines 48-50 please include a brief discussion of the recombinant protein used for these studies. Related, in Materials and Methods p. 13, please provide a few more details of protein construct, e.g. why is Cal tag used?

3. In Figure 1b and 1c, can the approximate location of the disordered VD be indicated in the ribbon diagram, perhaps as a yellow dashed line? Understood that its precise conformation is not known, but it would help to illustrate where this disordered loop is with respect to the stalks especially in the

dimer arrangement in 1c.

4. Figure 1a: can the size of the schematic be expanded a bit for easier reading
5. Please fix the order of the extended data according to their appearance. For example, Extended Data Fig 3d appears first in the results section.
6. Also related to Extended Data Fig 3d: what is the mass photometry described in the figure legend? It should be included in the Methods. Also, in the text line the concentration is given as 600 nM but the figure legend shows 100 nM. This should be addressed.
7. Please label GTPase domain helix α 2G (line 71) in Figure 1b.
8. Extended Data Fig 2d seems redundant with 2e in some of the sequences.
9. Lines 109-111, a reference of the published study should be used instead/in addition
10. Lines 110-111 please include residue numbering for Glutamine and Glutamic acid mutations from previous study.
11. Line 121 please add reference to Figure 2d when discussing CLnts negative stain results.
12. Line 124, are you referring to apo samples?
13. Reference to Fig 3c is missing in text.
14. Line 195: is the mutant incapable of assembly with itself, and/or with the CLnts?
15. Discussion of potential inhibitor design in light of the new structure: are there druggable sites?

Reviewer #2 (Remarks to the Author):

Rochon et al.

DRP1 is required for the fission of mitochondria. A number of structures, both using crystallography and cryoEM have informed our understanding of the domain arrangements, nucleotide binding and partner or membrane interactions of DRP1. While DRP1 is known to exist in a dimer-tetramer equilibrium, at concentrations used for structural biology, the tetramer has been the most appreciated form, where distinct dimeric interfaces contribute to the tetramer formation. The authors study the dimer only form of DRP1, which they obtain by using low DRP1 concentrations that supports dimeric species in abundance. They see some interesting features for this dimer. It has a more compact form than previously observed in crystal and cryoEM structures. They also find a BSE-G-domain lock mediated via loop L3s that may keep the protein in an auto-inhibited state, as well as a conserved tyrosine that may mediate the dimer interface. The authors report some interesting findings and I think the study is suitable for the audience of Nature Communications. I have a few suggestions:

Major:

1. The resolution of the maps just by inspection of the figures looks possibly worse than 5-6Å. The flexibility of the hinge regions of DRP1, without the stabilizing nucleotide and partner interactions could contribute to this. A lot of the authors' claims about residue-level information are based on comparisons with previous work and residues such as R456 and Y493 may not have proper side chain densities in their current structures. While their experiments suggest that they are on the right track, I believe that the study will be boosted by higher resolution in the stalk region, which I think is achievable. I highly recommend using masks for the stalk region and using RELION-based methods such as skipalign and/or symmetry expansion (presently the authors only use cryosparc). This will give the authors much more confidence in their claims. Mutating DRP1 residues and observing effects such as high/low GTP turnover and high/low sedimentation may happen because of allosteric effects and not directly because of the residues that they are targeting. This is not to say that they are not right, but a high resolution stalk region will really support their claims and make the paper stronger.
2. In the absence of high resolution information, could the authors engineer a Cys-Cys bond at the BSE lock (at R456) that they propose and show that even with the nucleotide, the locked DRP1 does not form spirals?
3. Kwapiszewska et al (Scientific reports, 2019) reported the presence of DRP1 dimers in cells. I think

this citation should be added to the body of the paper with the explanation for the common reader that dimers do exist at cellular DRP1 concentrations and what is being reported here is important.

4. In the map made available to me recently, I noticed large unassigned densities next to one of the G-domains (please see attached). Are these a mixture of structures with G-domain motion, perhaps? If so, using RELION-based classification is necessary to tease apart various sub-states and in the process increase the resolution of their maps, which is currently not very good, even in the stalks. For example, the even some stalk helices are not separated from one another, which would be expected of resolutions reported by the authors. I would estimate that the resolution is much worse than 5-6 angstroms as it stands. Please provide an independent resolution estimate as well, for example from Resmap, not just from cryosparc.

Minor:

1. The authors call their structure "native" at times. I would not use this term, since "native" is best suited for in situ structures. In my opinion dimeric DRP1 structure is more appropriate.
2. Line 91- the movement of the G domain is being compared to what? The legend for the Fig. 2a/b says both AlphaFold and crystal structure. On that note, why use AlphaFold at all, when so many experimental structures are available (and the authors make use of them).

RE: Manuscript NCOMMS-23-26691-T

We thank the reviewers for carefully reading the manuscript and offering suggestions that have led to a much-improved manuscript.

Reviewer #1 (Remarks to the Author):

In this study, Rochon et. al. present a cryo EM structure of full length Drp1 in a unique dimeric conformation, which is the authors interpret as representing an autoinhibited conformation since two previously established oligomerization interfaces are sterically blocked in this dimeric conformation. They identify two key conformational features that seem critical for this interpretation: the BSE lock and stalk interface. According to the introduction, it remained largely unknown how Drp1 and other dynamin proteins regulate their assembly into higher order oligomers when stimulating mitochondrial fission. This study attempts to fill this gap in knowledge by providing a view of an autoinhibited conformation. This conformation differs from previous crystal structures, and this new viewpoint may be of value to the dynamin field. Mutations at both sites were introduced to disrupt this autoinhibited conformation, and assessed in numerous ways including assembly in negative stain EM, enzymatic GAP assays, and transfection/microscopy studies. Taken together, these provide an experimentally comprehensive approach to the question, and only a few questions are raised below regarding the experimental data.

We appreciate the positive comments and have added an additional extended figure and additional illustrations to figure 4 to better demonstrate our interpretations based on the comments below. In addition, we are providing data and information to the reviewers for clarification. We are to make our responses to the reviewers available to the public if this manuscript is accepted for publication.

The studies seem generally technically sound, though I cannot comment directly on the details of the cryo EM due to limited expertise in this technique. The structural analysis, while performed appropriately, are unfortunately not consistently presented in a clear way that is easy to follow; I often found myself rereading sections in an attempt to grasp the intended analysis. I do believe that a careful reworking of some sections of the text, and/or some new figures or more clearly labeled panels, may allow this paper to appeal to a broader audience beyond just the specifics of dynamin structure; without this, I do not believe this study is quite ready for publication.

After careful review of the text, we have made edits to clarify our interpretations and to help the overall flow of the paper. Specific changes are outlined below.

The conclusions also seem overly drawn, with alternative explanations not always provided. While the provided data do not argue against the mechanism presented, this interpretation is not the only possible explanation. For example, the instances of “primed for assembly” versus observable assembly versus oligomerization were difficult to follow.

Overall, we were careful to not over-interpret the data due to limits in resolution of the structure(s). In the absence of side chain density, we used biochemistry to validate our interpretations of the structural insights. We do think we have successfully demonstrated an ability to alter the activity of Drp1 in a predictable way through targeted mutations.

We apologize for the confusion regarding the differences between an oligomer state versus an assembly state and have reworked the wording to better clarify the differences between assembly and multimerization. Specific examples are provided below.

Additionally, other than two GTPase assays, I do not believe that a strong connection between GTPase activity, GTPase domain conformation, and autoinhibitory/active conformational differences are explored enough. Does the G domain have the canonical switch regions, and are these observed in the cryo EM density? Is the interaction with the BSE lock dependent on the switch regions?

Yes, Drp1 has canonical switch regions. Similar to other DSPs, the binding affinity for nucleotide is much lower compared to other G proteins, so trying to capture different states is difficult as the nucleotide freely exchanges making distinct states transient. Within our structure, the GTP binding pocket of the protein is not as well resolved (it is unbound to nucleotide), making these interpretations more difficult. It would be interesting to suggest that the BSE lock can influence GTP binding, but we lack the resolution in the G1-G4 loops that would allow us to make reliable structural interpretations. While the resolution in the BSE region is quite good, this is far from the nucleotide binding site, so it is not obvious how the conformation in this regions would directly impact nucleotide binding and hydrolysis. Moreover, the binding pocket is not buried or occluded in our “open” and “closed” conformations. We can't rule out indirect effects, and this is an area of study we would like to pursue in the future.

Lastly, the abstract should be edited to include the findings beyond just the cryo EM structure.

We made some small changes to the abstract, but we were already at the word limit. The main conclusions are related to the implications of the structure.

SPECIFIC QUESTIONS/EDITS

1. Line 46. Can the authors please provide one or two additional sentences to frame the study; for example, to the current gap in the knowledge specifically for Drp1 and how the study directly addresses this?

We have added additional context to address this gap in knowledge.

Added at lines 45-50: This study sets out to resolve the structure of Drp1 in a full length, dimer state in solution. Drp1 is predominantly found in a dimer state at physiological conditions and is an important structure to resolve to understand mitochondrial division¹⁵. There exists no full-length sequence, solution structure of a DSP. This lack of structural information limits basic understanding of regulation, inhibition, and activation of Drp1 and other DSP fission proteins.

2. If 4 conformations were obtained, which one is the predominant structure being used for analysis? For example, Figure 1b. Some explanation of how this was chosen would be appropriate. Also, is the BSE lock discussed later on consistent in all these conformations?

Of the four structures, the differences were not extreme. In one conformation, a GTPase domain appeared to be in an extended conformation; however, it was the lowest quality map and we did not want to overstate our conclusions without additional confidence. The conformation chosen as the representative structure was populated by the most particles and was the most complete map. We have added additional explanation within the text for clarification.

Added at lines 62-68: Conformation 1 generated the highest quality map with the most particles, the best fit, and is used as the model dimer (Fig. 1b, c). In this map both BSEs are resolved, generating the most complete model with the highest confidence. The second G domain density is not fully resolved in any of the conformations. We believe this is due to the heterogeneity in the relative conformations of the G domains in the dimer. Conformations 2-4 provide insight into the heterogeneity in the dimer interface; however, resolution and map quality was better for conformation 1.

3. Lines 59-89. This whole section would benefit from more illustration. For example, how is larger assembly sterically blocked by the observed dimer conformation? Specifically, what is occluding the DSP helical structures that would otherwise mediate higher assembly; is the orientation of the GTPase domain? Is this true for all 4 conformations? This is a main conclusion of the study and yet it does not seem to be illustrated clearly.

We really appreciate this observation. We have added several panels to the final figure 4 and an additional extended figure 4 to illustrate the clashes that inhibit assembly. The figures are shown below and the following text were added to discuss the panels.

Added at lines 225-237: DSPs form helical assemblies through two highly conserved interfaces, interface 1 and interface 3 (Extended Data Fig. 4). Using dynamin 3 interfaces as a reference, we aligned two dimers of the Drp1 crystal structure and two dimers of the dimeric cryoEM structure to interface 1 (Fig. 4, a and b). The crystal structure tetramer aligned with no apparent clashes. Interface 3 was also found to be in a reasonable position relative to interface 1. When the dimeric cryoEM dimers were aligned to the dynamin 3 interface 1, it becomes apparent that the G domains are in a vastly different conformation compared to the crystal structure. Additionally, interface 3 is not aligned when interface 1 was the reference. To determine the possible clashes, interfaces 1 and 3 were aligned separately. In both alignments, obvious steric clashes were observed. Helices $\alpha 1C^S$ and $\alpha 1N^S$ and loops $L1N^S$ and $L2^S$ would prevent the dimeric cryoEM structure from forming either interface without undergoing conformation rearrangements. This is informative as mutations found in these two loops have been characterized to limit assembly.

Top panels of figure 4

New extended figure 4

4. Lines 61-66 discussion of hinge motion: Figure panels illustrating this should be explicitly stated in text. Similar problem with reading lines 77-89. Illustrations are needed.

Thank you for catching this. This section is meant to explain high level observations and we've removed any specific conclusions that could be drawn. We've added additional figure citations as well to direct the reader to panels that would be visually helpful.

5. Figure 1d it appears as though the GTPase density in chain B is weak/missing. Please add comments and interpretations of this in the text.

As stated above, the relative positions of the G domains led to heterogeneity in the cryo-EM map, which is likely why symmetry was not helpful in the reconstruction (as suggested by the other reviewer). We have added the text referenced in reviewer major comment 2 in the manuscript to clarify this observation.

6. Line 91, please explain which structures are being compared when saying "The G domain moves". From the Figure 2 legend, I see it is the crystal structure vs cryo EM structure, but what is the AlphaFold component being cited in the figure legend?

We thank the reviewer for catching this. The G domain is in a different conformation when comparing the model vs the cryo-EM structure. This is now changed in the manuscript to avoid confusion. Regarding the AlphaFold citation, the monomer generated in AlphaFold aligns completely to the crystal structure; however, it includes missing structure that was not resolved in the x-ray density. Specifically the loops. In figures where the approximate position of the loops are helpful, the AlphaFold structure was used.

Added lines 103-105: Comparing the autoinhibited dimer structure to the published crystal structure, the G domain of the cryo-EM structure is positioned 79 Å closer to the distal end of the stalk. This change in position is accompanied by 67° rotation and a 61° twist of the G domain (Fig. 2, a and b).

7. Line 115-, please explain the use of GMP-PCP in the experiment. Is the result meant to imply that the GTP-bound conformation of the G domain releases the autoinhibited state? Did you confirm that Drp1 in this sample is bound to GMP-PCP? Related, in the methods, please describe how exchange to GMP-PCP was performed and detected. Does the Drp1 purify with GDP bound?

The relative affinity for nucleotide is low for Drp1 and other DSPs (with a K_d in the μM range and K_m is over $100 \mu\text{M}^{1,2}$), so it is not expected to be in a nucleotide-bound state when purified. It only binds stably when nucleotide is added at saturating concentrations. The binding to GDP is weaker than GTP, so DSPs generally do not require a GEF. All experiments were carried out in saturating conditions of nucleotide (2 mM).

We used a non-hydrolysable nucleotide to characterize the propensity of protein and mutant constructs to assemble by locking it into rings or larger spiral assembly. The assembled state is different than large apo oligomers. We can confirm that the protein is bound to GMP-PCP through negative stain imaging. After the protein is purified, we incubate with an excess of GMP-PCP at room temperature. When Drp1 complexes with a nucleotide it forms rings and spirals. At the concentrations of protein used in this study, the dimers and tetramers (10-20 nm particles) are the most abundant multimers found in the sample, and larger species are exceedingly rare. When spirals and rings are observed, we know nucleotide binding promoted assembly because these structures are not observed in an apo state or in the absence of lipid.

Added lines 126-127: This use of a non-hydrolysable nucleotide locks the protein in a nucleotide-bound state.

8. Was cryo EM attempted on the R456E GMP-PCP protein to compare with the previous study of ref 4?

CryoEM was not attempted on the Drp1 R456E mutant complexed with GMP-PCP. The rings and spirals which form as a result of nucleotide binding are too heterogeneous and irregular to resolve high-resolution structural information. The Frost structure that we reference (ref. 4) was also determined in the presence with a partner protein which resulted in regular filaments. So this comparison would be complicated, and the efforts required to solve an additional cryo-EM structure would be substantial.

9. Fig 2d, 2 μM protein was used for negative stain grid. This is above the aforementioned 600 nM concentration used to enrich for dimers (line 51). Can you comment on the dimer versus tetramer in this sample, and whether this affects the results of the WT versus mutant?

It's been shown previously that Drp1 is in dimeric equilibrium³. At the concentrations used in all experiments (0.1 – 2 μM) the predominant oligomer state is expected to consistently be a dimer, and we wouldn't anticipate that this changes the structure. Physiological concentration is considered 500 nM⁴. When preparing grids for negative stain, it is common practice to use higher concentrations to promote sample adherence to the grid. We have done 2D class averages on samples of Drp1 in the absence of nucleotide at micromolar concentrations have

not found tetramers in the sample. The only time we have resolved tetramer class averages are in samples which were cross linked with glutaraldehyde.

10. Fig 2e. All data points should be shown overlaying bar graph. I am not convinced by the bar charts and error bars that the mutation is statistically different than WT in the PCP samples but not the CLnts. Also, the conclusion drawn is that release of the BSE lock is weakened with mutation, but that multimer formation is not affected; can this result be reconciled with the overall conclusion that the observed dimer is in the autoinhibited conformation? What is the difference between multimerization (not enhanced) versus assembly (which is "primed" in the mutant protein sample)?

The assembly into larger helical assemblies in the presence of nucleotide or a lipid template requires that the protein cycle down to a dimer based on previous studies. Specifically, mutations that enrich larger multimers (tetramer, hexamers, and so on) are not able to mediate fission². Additionally, the Drp1 dimer is the fundamental building block of larger polymers that have been studied using cryoEM^{5,6}.

Data points have been added to the bar charts. What we see is variability in the WT replicates while the sedimentation values for both mutants are more consistent.

11. Line 131-140. I am not sure that I agree with the interpretation that the fold increase of the mutant is functionally different than the WT. The mutant is still less active than the WT; is the mutant protein less stable? A small point: to aid in reading, please include in the text that the enzymatic activity being measured is the GTPase activity of the G domain using an endpoint malachite green assay.

We agree with this assessment and have revised the wording in this section.

Added the endpoint malachite green assay notation at *lines 149-150*.

Reworded at Lines 143-152: To complement the assembly assay, enzymatic activity was measured using an endpoint malachite green assay (Fig. 2h). R456E (0.32 min^{-1}) exhibited a five-fold decrease in basal activity compared WT (1.7 min^{-1}). This change likely reflects conformational differences in the protein since the multimer state in solution is not altered. The stimulated activity of R456E in the presence of CLnts (16.8 min^{-1}) was still lower compared to WT (35.1 min^{-1}), but there was still significant stimulation for the R456E when compared to its basal rate demonstrating its ability to form higher order oligomers. This is consistent with the assembly observed in the presence of GMP-PCP. Again, this mutant is more assembly-potent, and the lipid-bound structures look indistinguishable from WT (Fig. 2d), though we cannot discount small differences that affect this stimulation.

12. Lines 141-151 and Figure 2i-j. I have concerns with the methodology described regarding the blinded individuals scoring the mitochondrial phenotypes. If four categories were used, why are only two shown? How indistinguishable are the phenotypes from each other? Did multiple blinded individuals score the same datasets, and if so, what were the discrepancies across their scoring? The dataset seems quite small ($N = 20\text{s}$ to 40s only); was this N for each individual transfected cell, or transfected wells, or experiments? Is there a way to automatically score this phenotype? The bar graph reflects that the WT cells are neither fragmented nor fused; which category do the majority of WT MEF's belong to? CTCF is shown, but was this applied to the cells being scored?

Using a committee approach is a commonly used process due to the extreme variability of mitochondrial morphology across cell types. The committee participants scored the same data at the same time while blinded to the conditions. As the reviewer notes, the different phenotypes can be subtle (demonstrated in panel **a** below), which is why we have had more success with this method over an automated one. All categories were recorded though they were not all reported to simplify the figure, focusing only on the shift from fused to fragmented. We wanted to focus on the trend. The N represents the number of cells scored, as we make a call for the mitochondria pattern in each cell according to the most prevalent phenotype observed. The transfection efficiency was low and we didn't want to overpower the WT and EV data set. The WT cells are primarily intermediate phenotypes which is expected for this cell type. To highlight the complete distribution, we've added the bar charts with all phenotypes to extended figures 2 and 3 (shown below).

13. Extended Figure 2h needs labels. The WT cell (presumably on the left panel) looks fuzzy rather than diffuse. Extended Figure 2i, is the expression of R456E mutant higher? What might this mean for the observed phenotypes? Also, an anti-myc immunoblot should be performed to complement this CTCF and also confirm that the expressed proteins are full-length. Together with point 12 above, I feel the data are over interpreted with respect to the autoinhibited conformation hypothesis.

Labels were added, thanks for catching that mistake. The CTCF was done due to the low transfection efficiency of the KO MEFs. This analysis confirms expression was similar when comparing the WT to the mutants. Also due to the low efficiency, WBs exhibit a weak Drp1 signal with the anti-Myc or anti-Drp1 antibodies. The KO MEFs have been well characterized by the Sesaki lab; however, we do understand the concern regarding incomplete translation. We don't see incomplete translation products when reviewing the myc signal.

To confirm that these constructs express full-length Drp1, we transfected HeLa cells with the plasmid (shown below). There are complete translation products observed in this cell line with higher transfection efficiency rates.

14. Was a mutant in the corresponding surface on the alpha 2b considered?

We believe there are several possible residues involved in stabilizing this lock. We successfully mutated E324 with an arginine charge reversal. The cryoEM density supports a possible interaction with E324 and R456. However, E324 is in hinge 1 which is a critical hinge for assembly. The mutation resulted in an inactive protein, incapable of forming rings and spirals in the presence of GMP-PCP. More work will need to be done to identify the other residues involved in this interface. We believe it is a complex intramolecular interaction.

15. The description of the dimer interface would benefit from more illustrations and/or simpler text. As it stands, the text is confusing with respect to the heterogeneity and comparison with the previous crystal structures. The main point of this section, that Y493 is at the interface, is supported by weak density in Extended data Figure 3b; this panel should nonetheless be moved to the main figure due to its importance for setting up the remaining experiments.

We have moved that panel from the extended figure to the main figure 3.

16. I was surprised a modest Tyr to Ala mutation was enough to elicit this effect; what is known about the affinity of dimerization?

The buried surface area for the crystal structure was reported to be 1,000 Å² which is comprised of both hydrophobic interactions and salt bridges.⁷ These residues are highly conserved across DSPs. During the crystallographic studies, attempts were made to break the dimer interface. Two residues with charge reversals were found to disrupt the salt bridges bookending the interface (K642E and E490R). Interestingly, E490A had no effect. We believe Y493 is an important bookend that controls the range of motion while allowing flexibility in interface 2. Without the bulky tyrosine at this position, the flexibility may be exacerbated leading to disruption of this conserved interface.

17. Line 201: there is little evidence for the “shock absorber” mechanism. Can this be explained more clearly?

Lines 215-217, we changed the description to: Therefore, the Y493 residue is critical for stabilizing interface 2, bookending either side of the interface to accommodate flexibility from a continuous interface to a more discreet intermolecular interaction required for spiral and helical oligomerization.

MINOR Comments

1. Line 46, please provide references for crystallographic studies where the VD loop was removed.

Reference added (now at line 47 due to additional changes).

2. Lines 48-50 please include a brief discussion of the recombinant protein used for these studies. Related, in Materials and Methods p. 13, please provide a few more details of protein construct, e.g. why is Cal tag used?

For this study, we used Drp1 isoform 1 which includes a B-insert in the variable domain hoping the additional MW might make particle identification easier as this dimer is relatively small compared to most cryoEM structures. Additionally, studies have shown that this splice variant is more dimeric when compared to other isoforms². The construct with the calmodulin tag expresses well and has a specific affinity that yields clean protein, and our protein has similar attributes (activity, etc) when compared with other Drp1 purification protocols. We have added clarification in the manuscript.

Added at lines 54-56: For this study, WT Drp1 (isoform 1) was expressed in *E. coli*. This particular isoform is the second longest and includes the B-insert (exons 16 and 17). It was selected for the cryoEM experiments since it has previously been shown to more dimeric when compared with other splice variants².

3. In Figure 1b and 1c, can the approximate location of the disordered VD be indicated in the ribbon diagram, perhaps as a yellow dashed line? Understood that its precise conformation is not known, but it would help to illustrate where this disordered loop is with respect to the stalks especially in the dimer arrangement in 1c.

This has now been added.

4. Figure 1a: can the size of the schematic be expanded a bit for easier reading

The schematic has been expanded and the readability should be much improved.

5. Please fix the order of the extended data according to their appearance. For example, Extended Data Fig 3d appears first in the results section.

Extended data figure 3 references have been adjusted.

6. Also related to Extended Data Fig 3d: what is the mass photometry described in the figure legend? It should be included in the Methods. Also, in the text line the concentration is given as 600 nM but the figure legend shows 100 nM. This should be addressed.

The Mass Photometry is a light-scattering technique that measures the molecular mass of individual molecules in dilute solutions. The details of the instrument and experimental setup are described in the supplementary file as this method was used only for the extended data figure.

We appreciate the reviewer for finding this accidental omission. Similarly, the CTCF has been moved to the supplement as it is only found in the extended data.

7. Please label GTPase domain helix a2G (line 71) in Figure 1b.

This has been added to the figure.

8. Extended Data Fig 2d seems redundant with 2e in some of the sequences.

We agree and have removed 2d and modified 2e to highlight the structures being shown in 2c to remove the redundancy and confusion.

9. Lines 109-111, a reference of the published study should be used instead/in addition

We have removed this reference since it has not been published. It does not add to this manuscript.

10. Lines 110-111 please include residue numbering for Glutamine and Glutamic acid mutations from previous study.

Again, this has been removed to avoid confusion.

11. Line 121 please add reference to Figure 2d when discussing CLnts negative stain results.

Added figure reference at line 132

12. Line 124, are you referring to apo samples?

Added at lines 136-137: when not in the presence of a nucleotide

13. Reference to Fig 3c is missing in text

Thank you for catching this. This has now been added (line 189), and the letter changed to b due to changes to the order based on feedback.

14. Line 195: is the mutant incapable of assembly with itself, and/or with the CLnts?

This is a great question. We think it's both. It must form dimers in order to form a polymer around the NTs. It's inability to maintain a stable dimer state would lead to an inability to form larger assemblies.

Added clarification at lines 196-198: This is not surprising, considering the importance of the dimer. The dimer is Drp1's functional unit. If a stable dimer interface is interrupted, the protein will be incapable for forming larger assemblies.

15. Discussion of potential inhibitor design in light of the new structure: are there druggable sites?

Based on this study, we have not identified specific sites for drug targets. We believe this new understanding of the dimeric structure will aid groups in the development of new sites that may be targeted. The use of a small peptide or a small molecule at these critical interfaces may stabilize an assembly-incompetent state that would inhibit mitochondrial fission. This would be an exciting area for future studies.

Added at lines 286-290: These new insights will be important for developing new selective inhibitors of Drp1. To date, efforts to regulate Drp1 have been based on the open and active conformation of Drp1 and have failed to produce a reliable inhibitor targeting Drp1 directly. In the future, factors that stabilize regulatory interactions in the dimer state would limit assembly of the fission machinery and provide a novel approach to prevent mitochondrial fission.

Reviewer #2 (Remarks to the Author):

Rochon et al.

DRP1 is required for the fission of mitochondria. A number of structures, both using crystallography and cryoEM have informed our understanding of the domain arrangements, nucleotide binding and partner or membrane interactions of DRP1. While DRP1 is known to exist in a dimer-tetramer equilibrium, at concentrations used for structural biology, the tetramer has been the most appreciated form, where distinct dimeric interfaces contribute to the tetramer formation. The authors study the dimer only form of DRP1, which they obtain by using low DRP1 concentrations that supports dimeric species in abundance. They see some interesting features for this dimer. It has a more compact form than previously observed in crystal and cryoEM structures. They also find a BSE-G-domain lock mediated via loop L3s that may keep the protein in an auto-inhibited state, as well as a conserved tyrosine that may mediate the dimer interface. The authors report some interesting findings and I think the study is suitable for the audience of Nature Communications. I have a few suggestions:

Major:

1. The resolution of the maps just by inspection of the figures looks possibly worse than 5-6Å. The flexibility of the hinge regions of DRP1, without the stabilizing nucleotide and partner interactions could contribute to this. A lot of the authors' claims about residue-level information are based on comparisons with previous work and residues such as R456 and Y493 may not have proper side chain densities in their current structures. While their experiments suggest that they are on the right track, I believe that the study will be boosted by higher resolution in the stalk region, which I think is achievable.

We appreciate the questions regarding resolution. We understand there are areas of significant blurring or even missing density. We attempted to be very careful in the analysis of any residues, not making any claims as to the side positions due to the lack of resolution. Running ResMap resulting in higher resolution than cryoSPARC and we don't want to over report the quality of the map. This data was not included in the manuscript to avoid overestimating the quality of the map.

I highly recommend using masks for the stalk region and using RELION-based methods such as skipalign and/or symmetry expansion (presently the authors only use cryosparc). This will give the authors much more confidence in their claims. Mutating DRP1 residues and observing effects such as high/low GTP turnover and high/low sedimentation may happen because of allosteric effects and not directly because of the residues that they are targeting. This is not to say that they are not right, but a high resolution stalk region will really support their claims and make the paper stronger.

Efforts were made to use RELION as well as masking and symmetry expansion. The 2D class averages between the two platforms are comparable, giving us confidence in the particle stack. Unfortunately masking our protein to include just the stalks and the BSEs results in the removal over half the protein. We believe this results in increased noise despite any increase to the signal. Masks were also tried in combinations of just the stalks and both stalks with one BSE. None of these resulted in improved resolutions.

In addition, C2 symmetry and symmetry expansion was also attempted with no improvement to the map. The inherent flexibility of the protein was best resolved using the non-uniform refinement algorithm of cryoSPARC.

RELION 2D Classes

cryoSPARC 2D Classes

Mask Creation: Stalks + 2BSEs

cryoSPARC: Symmetry Expansion (C2)

cryoSPARC: mask + NU refine

2. In the absence of high resolution information, could the authors engineer a Cys-Cys bond at the BSE lock (at R456) that they propose and show that even with the nucleotide, the locked DRP1 does not form spirals?

We had some concerns about trying this because Drp1 has nine cystines. Attempts have been made to mutate all cystines to allow for such an experiment; however, these residues are critical for protein function (see MacDonald *et al.*, 2014, MBoC)³. Because of these cystines, we know that the protein is sensitive to oxidation and reducing agents are required in our experimental buffers. Regardless, we may a double Cys mutation (E324C and R456C) and gentle oxidation was performed. In any conditions that we tested, we found that oxidation of the WT protein rendered the protein incapable of forming spirals or functional oligomers (almost certainly due to oxidation of natural cysteines), so any defect observed with the double Cys was indistinguishable from WT defects.

3. Kwapiszewska et al (Scientific reports, 2019) reported the presence of DRP1 dimers in cells. I think this citation should be added to the body of the paper with the explanation for the common reader that dimers do exist at cellular DRP1 concentrations and what is being reported here is important.

Thank you for calling attention to this publication. The importance of this structure was emphasized with the following citation.

Added at lines 45-50: This study sets out to resolve the structure of Drp1 in a full length, dimer state in solution. Drp1 is predominantly found in a dimer state at physiological conditions and is an important structure to resolve to understand mitochondrial division¹⁵. There exists no full-length sequence, solution structure of a DSP. This lack of structural information limits basic understanding of regulation, inhibition, and activation of Drp1 and other DSP fission proteins.

4. In the map made available to me recently, I noticed large unassigned densities next to one of the G-domains (please see attached). Are these a mixture of structures with G-domain motion, perhaps?

We think this is probably correct. The flexibility conferred to the G domains through hinge one and the BSE hinge results in different G domain positions relative to one another. We think the refine and classification jobs are able to resolve one at the expense of the other.

Added at lines 62-68: Conformation 1 generated the highest quality map with the most particles, the best fit, and is used as the model dimer (Fig. 1b, c). In this map both BSEs are resolved, generating the most complete model with the highest confidence. The second G domain density is not fully resolved in any of the conformations. We believe this is due to the heterogeneity in the relative conformations of the G domains in the dimer. Conformations 2-4 provide insight into the heterogeneity in the dimer interface; however, resolution and map quality was better for conformation 1.

If so, using RELION-based classification is necessary to tease apart various sub-states and in the process increase the resolution of their maps, which is currently not very good, even in the stalks. For example, the even some stalk helices are not separated from one another, which would be expected of resolutions reported by the authors. I would estimate that the resolution is much worse than 5-6 angstroms as it stands. Please provide an independent resolution estimate as well, for example from Resmap, not just from cryosparc.

We believe the problem here is twofold. In addition to the heterogeneity, the protein does have preferred orientations as evidence by the 2D classification. Even rebalancing the particles did not tease out the missing angles. The lack of separation observed from some views may reflect this limitation.

Minor comments:

1. The authors call their structure “native” at times. I would not use this term, since “native” is best suited for in situ structures. In my opinion dimeric DRP1 structure is more appropriate.

Agreed. We have changed all references of native to either dimeric or full-length.

2. Line 91- the movement of the G domain is being compared to what? The legend for the Fig.

2a/b says both AlphaFold and crystal structure. On that note, why use AlphaFold at all, when so many experimental structures are available (and the authors make use of them).

This was noted by other reviewer, and it is not a movement. We have changed the wording to highlight the difference in positioning of the domains relative to one another in the different structures.

Added lines 103-105: Comparing the autoinhibited dimer structure to the published crystal structure, the G domain of the cryo-EM structure is positioned 79 Å closer to the distal end of the stalk. This change in position is accompanied by 67° rotation and a 61° twist of the G domain (Fig. 2, a and b).

References

- 1 Macdonald, P. J. *et al.* Distinct Splice Variants of Dynamin-related Protein 1 Differentially Utilize Mitochondrial Fission Factor as an Effector of Cooperative GTPase Activity. *J Biol Chem* **291**, 493-507 (2016). <https://doi.org/10.1074/jbc.M115.680181>
- 2 Lu, B. *et al.* Steric interference from intrinsically disordered regions controls dynamin-related protein 1 self-assembly during mitochondrial fission. *Sci Rep* **8**, 10879 (2018). <https://doi.org/10.1038/s41598-018-29001-9>
- 3 Macdonald, P. J. *et al.* A dimeric equilibrium intermediate nucleates Drp1 reassembly on mitochondrial membranes for fission. *Molecular biology of the cell* **25**, 1905-1915 (2014). <https://doi.org/10.1091/mbc.E14-02-0728> [doi]
- 4 Hatch, A. L., Ji, W. K., Merrill, R. A., Strack, S. & Higgs, H. N. Actin filaments as dynamic reservoirs for Drp1 recruitment. *Mol Biol Cell* **27**, 3109-3121 (2016). <https://doi.org/10.1091/mbc.E16-03-0193>
- 5 Francy, C. A., Clinton, R. W., Frohlich, C., Murphy, C. & Mears, J. A. Cryo-EM Studies of Drp1 Reveal Cardiolipin Interactions that Activate the Helical Oligomer. *Scientific reports* **7**, 3 (2017). <https://doi.org/10.1038/s41598-017-11008-3> [doi]
- 6 Kalia, R. *et al.* Structural basis of mitochondrial receptor binding and constriction by DRP1. *Nature* **558**, 401-405 (2018). <https://doi.org/10.1038/s41586-018-0211-2>
- 7 Fröhlich, C. *et al.* Structural insights into oligomerization and mitochondrial remodelling of dynamin 1-like protein. *The EMBO Journal* **32**, 1280-1292 (2013). <https://doi.org/10.1038/emboj.2013.74>

Reviewer #1 (Remarks to the Author):

The authors have satisfactorily responded to my queries and have addressed most of my concerns in the revised manuscript. I therefore find this current study suitable for publication.

Reviewer #2 (Remarks to the Author):

I have reviewed the updated manuscript where authors describe a dimeric DRP1 structure. As before, there are noteworthy new things to learn from the structure, including the compaction seen in the dimer, the inhibition for oligomeric assembly in the compacted molecule and some key residues that may be mediating this structural state. I like the manuscript, but I still have concerns regarding the resolution of their map. I understand that given the low resolution, they have done biochemical and functional experiments to test their hypotheses, but the fact that the map resolution continues to be poor (especially in the stalk region where it is expected to be better) and that they source their mutations to this region makes me want to suggest one or two additional ways to make the map better.

It looks like there is a severe preferred orientation that this molecule has on cryoEM grids. Have the authors tried anything other than the GO grids that they mention? How about just holey carbon quantifoils? Or thin carbon-coated grids? Given the fact that they have the protein preparation presumably ready to go, this should not be difficult. Also, orientation could be influenced by adding glycerol or sub-cmc amounts of detergent, especially octylglucoside or even by tilting the microscope stage and collecting data at +/- 30deg or so. I believe that this is worth the effort to improve map quality, which does not look good currently.

In case none of the approaches ever work to improve map quality, then my suggestion is that the authors low pass filter their maps to 7-8 angstroms, and make figures with those, which shall smoothen the map. They should then also explicitly say that the resolutions claimed between 5-6A are not really evident visually from the map, and their mutants are thus a best possible measure of testing their model. This way, they show better looking maps, not over-sharpened stalks. My suggestion, however, is to improve the maps. From our work with DRP1, we know that mutations can affect DRP1 activity allosterically so it is always better to get side chain densities before claiming that a particular mutant directly leads to a particular effect.

Minor points:

1. Sedimentation assays as done in Fig. 2f are really not that informative, since both wild type and mutant DRP1 proteins sediment. Could the authors instead represent the proteins by their propensity to form rings, assembled ring stacks or no polymers? These data are already available to them from their negative stain and are probably more informative while comparing mutants vs wild type. Sedimentation assays look better when there is a big difference in the sedimentation properties between things being tested.
2. Extended data fig 4. PDB 5WP9 is the DRP1-MID49 co-polymer structure, not the Dyn3 crystal structure.

REVIEWER COMMENTS

Reviewer #1 (Remarks to the Author):

The authors have satisfactorily responded to my queries and have addressed most of my concerns in the revised manuscript. I therefore find this current study suitable for publication.

We thank the reviewer for their helpful comments that led to an improved manuscript.

Reviewer #2 (Remarks to the Author):

I have reviewed the updated manuscript where authors describe a dimeric DRP1 structure. As before, there are noteworthy new things to learn from the structure, including the compaction seen in the dimer, the inhibition for oligomeric assembly in the compacted molecule and some key residues that may be mediating this structural state. I like the manuscript, but I still have concerns regarding the resolution of their map. I understand that given the low resolution, they have done biochemical and functional experiments to test their hypotheses, but the fact that the map resolution continues to be poor (especially in the stalk region where it is expected to be better) and that they source their mutations to this region makes me want to suggest one or two additional ways to make the map better.

It looks like there is a severe preferred orientation that this molecule has on cryoEM grids. Have the authors tried anything other than the GO grids that they mention? How about just holey carbon quantifoils? Or thin carbon-coated grids? Given the fact that they have the protein preparation presumably ready to go, this should not be difficult. Also, orientation could be influenced by adding glycerol or sub-cmc amounts of detergent, especially octylglucoside or even by tilting the microscope stage and collecting data at +/- 30deg or so. I believe that this is worth the effort to improve map quality, which does not look good currently.

We thank the reviewer for these suggestions. We have tried many different strategies with sample preparation and had hoped to achieve a high-resolution structure as well. There are multiple factors that may contribute to the current limitations, including preferred orientation. With the current manuscript, the GO grids yielded the best structures despite using multiple strategies for data collection, but the resolution is still limited. We have not been able to resolve the preferred orientation, even with holey carbon grids. While we also expected that the stalk interface would have the highest resolution, there is considerable flexion around the core interface (Interface 2) that is evident even with the 3D classification. We do see value in trying additional preparation strategies, and sample optimization will continue in the lab (we appreciate some of the strategies suggested). However, implementing these other strategies represents a significant effort that will require considerable time to evaluate. In our view, the major conformational rearrangements that we highlight in the dimer structure are still evident at moderate resolution, and the interpretation is not altered.

We provide some examples below with the holey carbon, and the preferred orientation persists. In addition to data collections below, we've also tried 2D Class Rebalancing and cryoDRGN to overcome some of the issues with the preferred orientation and sample heterogeneity. Neither approach resulted in a better map.

Additional data collection results are presented below. We were still unable to capture some of the views. Resolution was even more limited with these samples, and the alignments never resolved high-resolution features despite collection of large datasets.

1) Drp1 samples on Quantifoil 2/1 holey carbon WITHOUT graphene oxide collected at SLAC.
Highest resolution: 14-16Å

2) Drp1 sample applied to grids without a graphene oxide support using Chameleon (blotless) application and collected at NCCAT.
Highest resolution: 12-15Å

3) GraFixed WT dimer fractions applied to Quantifoil holey carbon grids and collected on Titan Krios.
Highest resolution: 16-18Å

Additional data processing results are presented below.

4) 2D Class Average rebalancing

Distribution of current map:

Rebalancing resulted in poorer map qualities with little improvement in angular sampling:

In case none of the approaches ever work to improve map quality, then my suggestion is that the authors low pass filter their maps to 7-8 angstroms, and make figures with those, which shall smoothen the map. They should then also explicitly say that the resolutions claimed between 5-6Å are not really evident visually from the map, and their mutants are thus a best possible measure of testing their model. This way, they show better looking maps, not over-sharpened stalks. My suggestion, however, is to improve the maps. From our work with DRP1, we know that mutations can affect DRP1 activity allosterically so it is always better to get side chain densities before claiming that a particular mutant directly leads to a particular effect.

Based on this suggestion (and the lack of improvement based on other methods that we've tested), we have applied a low pass filter of 6 Å to the map in the manuscript. We did not sharpen the map. We tried filtering at 6, 7 and 8 Å resolution, and we did not see any apparent difference (see below), so we went with the 6 Å filter to be more consistent with the FSC. This does smoothen the map as suggested.

We also clearly state throughout the manuscript that there were no side chains evident in the map, and we were limited to secondary structure elements with our fitting. We have also made clear in the text that we're not claiming a global resolution if 5-6 Å. Every representation of the structure is without sidechains, and we are careful to point out that the mutations were designed with consideration of sequence conservation in the region of interest. As noted, these mutants represent the best way to test our model.

Updated lines 59-66: Using cryo-EM, the structure of a full-length dimer of WT human Drp1 was resolved to a reported resolution of 5.97 Å, which prevents the identification of side chains but secondary structure can be observed in regions of the map (Table 1, Extended Data Fig. 1). Four conformations were identified with significant conformational heterogeneity conferred

primarily through GTPase motions that highlight variability in the position of this domain relative to the stalk. Conformation 1 generated the highest quality map with the most particles, the best fit, and is used as the model dimer after applying a low pass filter of 6 Å to remove over-fitting artifacts (Fig. 1b, c).

Minor points:

1. Sedimentation assays as done in Fig. 2f are really not that informative, since both wild type and mutant DRP1 proteins sediment. Could the authors instead represent the proteins by their propensity to form rings, assembled ring stacks or no polymers? These data are already available to them from their negative stain and are probably more informative while comparing mutants vs wild type. Sedimentation assays look better when there is a big difference in the sedimentation properties between things being tested.

We have added three panels to extended data figure 2 which demonstrates this shift. The WT protein forms predominantly rings, while the R456E protein is found to form long spirals with very few rings present.

Added to lines 131-134: Negative stain images show that after two hours of incubation, WT forms predominantly rings with short spirals averaging .01 μm in length while R456E was observed to have fewer rings and the observed spirals were extended in length (2.6 fold increase compared to WT; Extended Fig. 2g-i).

2. Extended data fig 4. PDB 5WP9 is the DRP1-MID49 co-polymer structure, not the Dyn3 crystal structure.

Thank you for catching this error. It is the Dynamin 3 tetramer and the PBD ID was misattributed. It has been changed to 5A3F. We chose this structure as the model with a sequence alignment as this structure has identified residues for interfaces 1 and 3.

Reviewer #2 (Remarks to the Author):

I have reviewed the manuscript re-submission. I have the following small suggestions:

1. Please include a supplemental figure with a map-to-model FSC curve with the reported resolution at 0.5, which shall provide another angle to the reader to understand the overall resolution.

2. Please also include a directional FSC curve for evaluating directional resolution:

<https://www.ncbi.nlm.nih.gov/pmc/articles/PMC8294179/>

Both should be straightforward to do, and will add to the understanding of orientation bias and directional resolution for the sample.

With this, I am on board with publication. I am also of the opinion, that the resolution of this sample should definitely be improved with future studies. There may still be a lot to learn with high resolution maps.

REVIEWERS' COMMENTS

Reviewer #2 (Remarks to the Author):

I have reviewed the manuscript re-submission. I have the following small suggestions:

1. Please include a supplemental figure with a map-to-model FSC curve with the reported resolution at 0.5, which shall provide another angle to the reader to understand the overall resolution.

The 0.5 threshold has been added to the FSC curves in the workflow panel.

2. Please also include a directional FSC curve for evaluating directional resolution: <https://www.ncbi.nlm.nih.gov/pmc/articles/PMC8294179/>
Both should be straightforward to do, and will add to the understanding of orientation bias and directional resolution for the sample.

The 3DFSC has been added to supplemental figure 1 with the directional data to demonstrate further the preferred orientations.

With this, I am on board with publication. I am also of the opinion, that the resolution of this sample should definitely be improved with future studies. There may still be a lot to learn with high resolution maps.

We thank the reviewer for their suggestions regarding the resolution challenges. We're excited by the observations of this data and understand there are future efforts which can improve these results.